# Confidence drives a neural confirmation bias

Max Rollwage [1,2✉], Alisa Loosen [1,2], Tobias U. Hauser [1,2], Rani Moran[1,2], Raymond J. Dolan[1,2] & Stephen M. Fleming [1,2,3]

A prominent source of polarised and entrenched beliefs is confirmation bias, where evidence against one's position is selectively disregarded. This effect is most starkly evident when opposing parties are highly confident in their decisions. Here we combine human magnetoencephalography (MEG) with behavioural and neural modelling to identify alterations in post-decisional processing that contribute to the phenomenon of confirmation bias. We show that holding high confidence in a decision leads to a striking modulation of post-decision neural processing, such that integration of confirmatory evidence is amplified while disconfirmatory evidence processing is abolished. We conclude that confidence shapes a selective neural gating for choice-consistent information, reducing the likelihood of changes of mind on the basis of new information. A central role for confidence in shaping the fidelity of evidence accumulation indicates that metacognitive interventions may help ameliorate this pervasive cognitive bias.

[1] Wellcome Centre for Human Neuroimaging, University College London, London WC1N 3BG, UK. [2] Max Planck University College London Centre for Computational Psychiatry and Ageing Research, London WC1B 5EH, UK. [3] Department of Experimental Psychology, University College London, London WC1H 0AP, UK. ✉email: max.rollwage.16@ucl.ac.uk

The philosopher Bertrand Russell opined "The most savage controversies are about matters as to which there is no good evidence either way". While this view applies in some situations, even more troubling are instances where polarization and entrenchment of opinion persists in the face of contrary evidence, exemplified by debates on climate change and vaccinations. This polarization is most evident when opposing parties are highly confident in their positions[1,2]. A psychological-level explanation for such entrenchment is the idea that people selectively incorporate evidence in line with their beliefs, known as confirmation bias[3]. Although an extensive literature has documented this bias in behaviour[3,4], the underlying cognitive, computational and neuronal mechanisms are not understood.

So far, an investigation of confirmation bias has been restricted largely to scenarios involving complex real-world beliefs such as political attitudes[4–6]. However, the complexity of such higher-order beliefs makes it difficult to disentangle the various contributors to biased information processing. For instance, people may have a strong personal investment in their political opinions, leading to a significant motivation to discount new information that goes against their beliefs. Intriguingly, confirmation biases have recently been demonstrated in low-level perceptual tasks[7–9], that are unlikely to evoke such motivated reasoning. These studies indicate a source of confirmation bias may be a generic shift in the way the brain incorporates new information. Here we adopt such a task to study the computational and neural basis of post-decisional shifts in sensitivity to choice-consistent information.

Perceptual decision-making is well-described using sequential sampling models which assume the brain accumulates noisy evidence for each choice option to a decision bound[10]. This accumulation process is thought to be supported by neuronal populations in parietal and prefrontal cortex[11,12]. Importantly, while perceptual tasks allow tight control over the processes involved, they also permit generalisation to more complex decisions[13–15], and similar principles appear to underlie choice and confidence formation in both simple and more complex tasks[16,17]. However while the processes underlying perceptual decision-making have been studied in detail, little is known about the mechanisms governing accumulation of evidence after a choice has been made, or how such processing is shaped by pre-existing beliefs and confidence[7,17–23].

Here we combine theoretical models and neural metrics to identify alterations in post-decisional processing that may contribute to the phenomenon of confirmation bias. Across all experiments, participants were presented with a sample of moving dots (pre-decision evidence) before indicating their initial decision (motion to the left or right) and confidence in their choice (see Fig. 1a). They were then presented with a second sample of moving dots (post-decision evidence) before making a final choice and providing a confidence estimate. Importantly, pre- and post-decision evidence always indicated the same direction of motion such that post-decision evidence was helpful. Accordingly, an ideal Bayesian observer should use post-decision evidence to change its mind after initial mistakes (see Supplementary Note 1 for analysis of the adaptive usage of post-decision evidence), whereas a confirmation bias would blunt this belief flexibility[13,18].

## Results

### Effects of confidence on changes of mind

In a first experiment we hypothesised that a confirmation bias would occur more often when people are highly confident in their original choice[24–26]. In order to dissociate subjective confidence from objective performance we used a psychophysical manipulation ("positive

evidence"[27], see Methods) to selectively boost participants' confidence (mean difference = 0.024, CI = [0.008, 0.04], Cohen's $d$ = 0.21, $t(27)$ = 3.0, $p$ = 0.005, Fig. 1c) while leaving performance (mean difference = 0.006, CI = [−0.022, 0.034], Cohen's $d$ = 0.02; Bayesian $t$-test indicating equality: $BF_{01}$ = 4.61; Fig. 1b) and reaction times (mean difference = −0.005, CI = [−0.029, 0.018], Cohen's $d$ = −0.04; Bayesian $t$-test indicating equality: $BF_{01}$ = 4.51, Supplementary Fig. 5a) unaffected.

We next set out to test whether this boost in confidence influenced changes of mind. There were notable individual differences in the degree to which our manipulation boosted participants' confidence (see Fig. 1c, d). Importantly, subjects who experienced a stronger confidence boost through the positive evidence manipulation also showed a stronger reduction in changes of mind ($r$ = −0.69, $p$ < 0.0001, see Fig. 1d), an effect not explained by an impact of positive evidence on accuracy or reaction time ($p$ = 0.005 when controlling for these effects). This supports a notion that confidence drives reductions in changes of mind (see Supplementary Notes 5 and 6 for additional behavioural and magnetoencephalography (MEG) analyses that further confirm confidence as a critical driver of changes of mind).

### Confidence induces a selective gain for confirmatory evidence

We next reasoned that confidence may reduce changes of mind by promoting a bias towards processing of confirmatory post-decision evidence. We sought to test this hypothesis by revealing the process through which confidence affects accumulation of post-decision evidence, applying a combination of drift-diffusion modelling and recordings of post-decisional fluctuations in a neural decision variable (DV) using MEG. We considered two potential mechanisms through which confidence might reduce changes of mind. First, confidence might reflect a shift in the starting point of post-decision accumulation to be closer to the bound associated with an initial decision, consistent with a continuation of pre-decisional evidence accumulation (influence on starting point; Fig. 2a upper panel). Second, confidence may induce selective accumulation of evidence in line with an initial decision (influence on drift rate; Fig. 2a lower panel)—a clear instance of confirmation bias.

Critically, these two mechanisms make different predictions in terms of the distributions of response times for the final decision[8,9]. We compared 10 drift-diffusion models (DDMs) that embodied these different predictions (see Supplementary Note 2 for a full model comparison). We employed accuracy coding such that the bounds correspond to a correct versus an incorrect decision, such that a positive drift-rate represents stronger integration of the presented (correct) motion direction. Note, by design, confirmatory post-decision evidence was received when an initial decision was correct, and disconfirmatory evidence when an initial decision was incorrect (Fig. 2b–d). In addition, in light of suggestions that confidence might also affect the separation of decision bounds, and thus the trade-off between speed and accuracy of subsequent decisions[28,29] we allowed for a dependency of boundary separation on initial confidence in all models.

The models differed as to whether the starting point and/or drift-rate were affected by confidence (models 2–4), accuracy of the initial decision (models 5–7; i.e. correct = 1 and incorrect = −1, capturing a general confirmation bias) and their interaction (models 8–10; i.e. capturing a confirmation bias that depends on confidence). The winning model (Model 10, as indicated by the Deviance Information Criterion score; see Supplementary Fig. 2A) incorporated dependencies of starting point and drift-rate on all factors (confidence, initial decision and the interaction) and

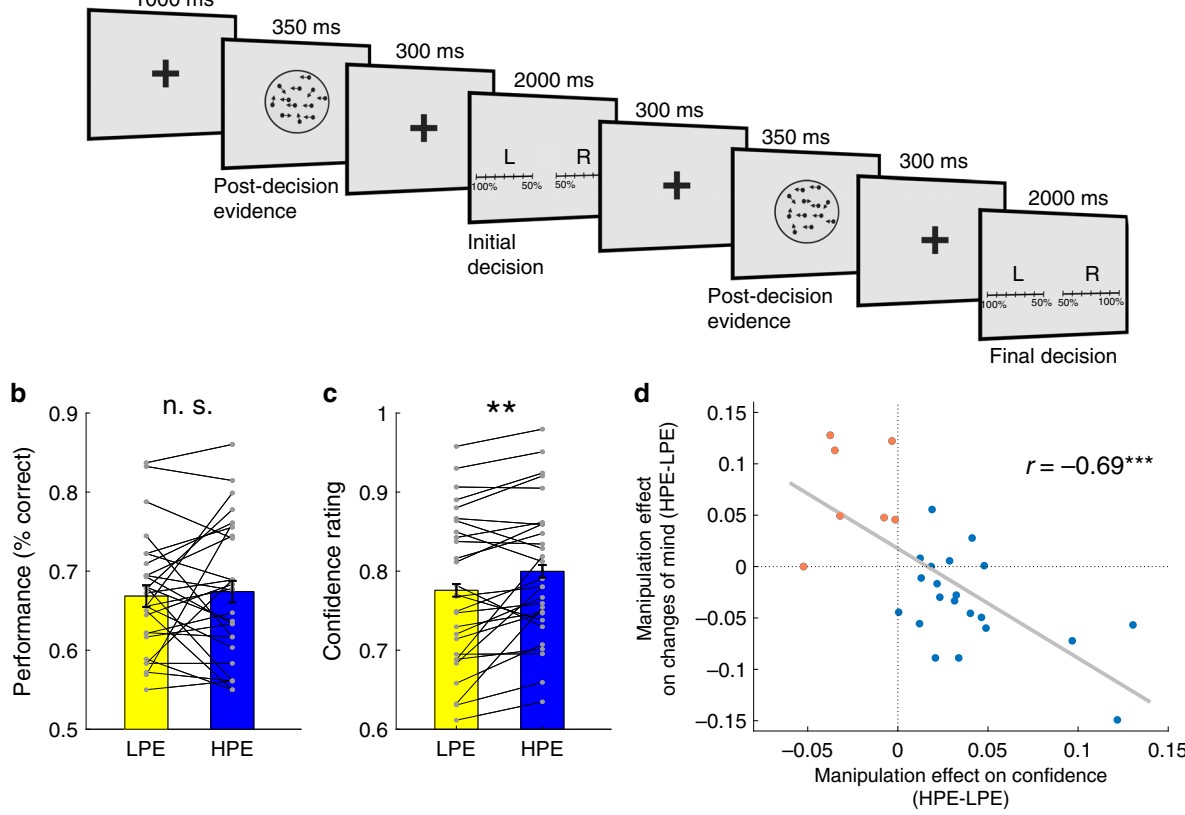

**Fig. 1 Task design and results of behavioural study 1 (_n_ = 28 participants). a** Trial timeline. Note that participants first had to indicate a binary left versus right decision (i.e. a two alternative forced-choice), and then indicate their confidence in this decision by moving a cursor along the selected scale. **b, c** A psychophysical manipulation of positive evidence selectively increased confidence of the first decision (**c**) while keeping accuracy constant (**b**). This increase in confidence was replicated across all three studies. Data are presented as mean values ± SEM; grey dots represent individual participant data. Paired _t_-test (two-tailed): **\*\*_p_ = .005. LPE = low positive evidence condition; HPE = high positive evidence condition. **d** Between-subject relationship between the degree to which positive evidence increased confidence (_x_-axis: confidence in the high positive evidence condition—confidence in the low positive evidence condition) and its effect on changes of mind (_y_-axis: changes of mind in the high positive evidence condition—changes of mind in the low positive evidence condition). This correlation was replicated in all three studies. Orange data points represent subjects showing the opposite of the intended effect of the manipulation on confidence (higher confidence in the low positive evidence condition). Pearson correlation (two-tailed): **\*\*\*_p_ < .0001.

provided a good fit to the data (Fig. 2b, c):

$$\text{Starting point} \sim 1 + \text{confidence} + \text{initial decision} \\ + \text{confidence} \times \text{initial decision} \tag{1}$$

$$\text{Drift-rate} \sim 1 + \text{post-decision evidence strength} \\ + \text{confidence} + \text{initial decision} + \text{confidence} \\ \times \text{initial decision} \tag{2}$$

$$\text{Boundary separation} \sim 1 + \text{confidence} \tag{3}$$

After accounting for main effects, we observed a dependency of the starting point on the interaction between confidence and initial decision (95% equal-tailed interval = 0.08−0.18; Fig. 2d right hand panel), indicating participants started the accumulation process closer to the bound of the initial decision when highly confident in their choice. Even more striking was the discovery of a similar interaction effect on drift rate (95% equal-tailed interval = 0.11−0.26; Fig. 2d right hand panel) indicating participants selectively accumulated evidence supporting their initial choice, and were more likely to do so when they were more confident. While a confidence-related shift in starting point might reflect normative usage of pre-decision evidence (because high confidence in an initial decision might reflect greater pre-decision

evidence accumulation, and thus be closer to a post-decisional bound), an influence of confidence on the drift rate is a clear instance of confirmation bias. Indeed, effects of the initial decision and confidence on the drift rate were more pronounced than those on the starting point (see Fig. 2 and Supplementary Note 3). Such a confirmation bias led to a boost in accumulation of the veridical motion direction following high-confidence correct decisions (as such information served to confirm the original choice), whereas it led to a reduction in evidence accumulation (manifest as a lowered drift rate) following high-confidence errors (as new information served to disconfirm an originally wrong decision).

**Neural markers of post-decisional processing.** While our DDM fits support a distinct influence of initial choice and confidence on post-decisional processing, they allow only indirect inference on how confidence affects evidence accumulation. To quantify this process more directly we used MEG to obtain a time-resolved neural metric of post-decision accumulation. Specifically, we trained a SVM classifier on brain activity (normalized amplitude of all MEG channels) at each time point (10 ms timebins) in the pre-decision time window (lasting 850 ms from stimulus onset to the presentation of choice options; note that the trial timeline for the MEG study differed slightly to the timeline presented in Fig. 1a, see "Methods" for details) to predict which choice (left or

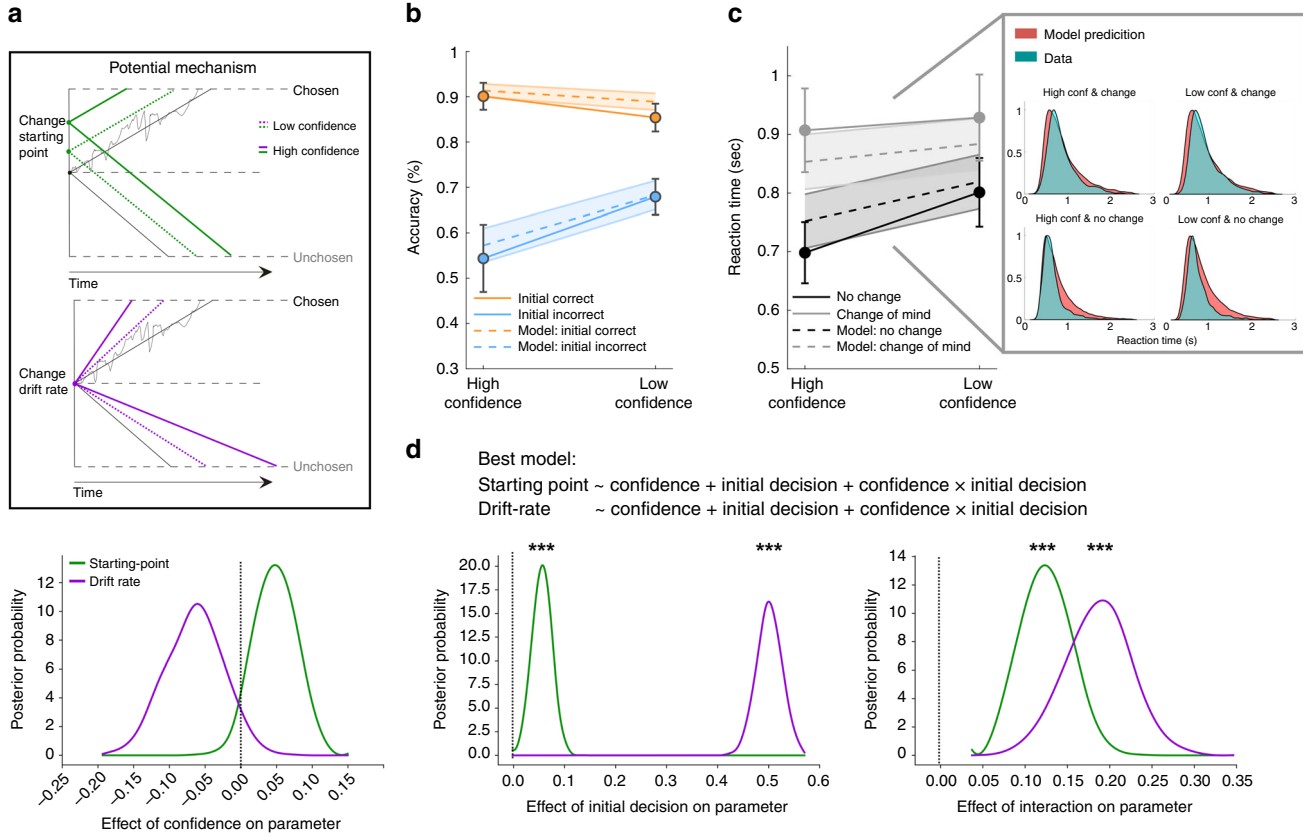

**Fig. 2 Drift-diffusion modelling fits to the second decision (behavioural study 2, n = 23 participants). a** Illustration of how confidence may reduce changes of mind through either a shift in starting point towards the decision bound of the initial decision (upper panel) and/or a selective increase of drift-rate for evidence supporting the initial decision (lower panel). **b, c** Model simulations (of the best fitting model) reproduce behavioural patterns of accuracy and reaction times of the second decision when plotted as a function of the initial decision and initial confidence. Due to the task structure participants received confirming post-decision evidence when they were initially correct and disconfirming post-decision evidence after initial mistakes. Model simulations are shown as dotted lines, behavioural data as solid lines. Data are presented as mean values +/− 95% confidence intervals. The righthand panel of (**c**) plots the full distribution of response times and model predictions for the different trial types (high confidence and no change of mind, low confidence and no change of mind, high confidence and change of mind, low confidence and change of mind). **d** Posterior distribution of model parameters of the best-fitting model. The dependencies of the drift rate (purple lines) and starting point (green lines) on initial confidence (left panel), initial decision (middle panel) and the interaction between confidence × initial decision (right panel) are presented. The dotted vertical lines represent an effect of zero/no effect. Note that these dependencies are simultaneously fitted, controlling for mutual influences. Markov-Chain Monte-Carlo sampling of posterior parameter distribution: ***$P$(parameter > 0)>0.999. Sec=seconds.

right) was made on each trial. We then applied the trained classifier to brain activity at the corresponding time point in the post-decision time window, enabling us to derive a probabilistic prediction of neural evidence favouring a leftward versus rightward decision (see Fig. 3a left panel). Positive values indicate prediction of a rightward decision and negative values indicate prediction of a leftward decision (see Fig. 3b). We next fitted a linear regression to the time series of classifier predictions within each trial (see Fig. 3a right panel) to obtain a trial-by-trial neural measure of the starting point (intercept) and drift rate (slope). These measures of neural evidence accumulation (slope) should be highly responsive to the presented motion direction during the post-decision period, and we show this was indeed the case (hierarchical regression: $\beta = 0.07$, $t(8550) = 6.89$, $p < 10^{-11}$, Fig. 3b).

The slopes extracted from this analysis are signed, such that positive values indicate evidence for a rightward choice and negative values evidence for a leftward choice. In order to obtain an unsigned metric of evidence accumulation strength, we flipped the sign of slopes extracted from trials in which leftward motion was presented (we conducted the same flip for the intercept to obtain an unsigned metric of the starting point). This unsigned metric quantifies a propensity to correctly integrate the presented

information, analogous to a drift rate in the accuracy coded DDM employed in Fig. 2.

A neural analogue of the drift-rate (or change in internal DV) should be related to characteristic features of the observer's decision. Specifically, stronger internal evidence accumulation should be related to a higher likelihood of having made a correct decision[12], faster response times[10] and higher confidence[11]. In order to check whether our classifier predictions satisfied these criteria for metrics of internal evidence accumulation, we entered both the trial-by-trial slope and intercept of the post-decision accumulation process as simultaneous predictors in a hierarchical regression model to predict (a) reaction times, (b) choice accuracy and (c) confidence of the final decision (see Supplementary Note 4 for a similar analysis of the pre-decision period). Steeper slopes predicted faster reaction times ($\beta = -0.007$, $t(8549) = -2.83$, $p = 0.005$, see Fig. 3d), a higher likelihood of a correct decision ($\beta = 0.16$, $t(8549) = 3.05$, $p = 0.002$, see Fig. 3e) and higher confidence ($\beta = 0.14$, $t(8549) = 3.53$, $p = 0.0004$, see Fig. 3f). We also observed significant effects of the intercept on accuracy ($\beta = 0.1$, $t(8549) = 2.0$, $p = 0.045$, see Fig. 3e) and confidence ($\beta = 0.12$, $t(8549) = 3.07$, $p = 0.002$, see Fig. 3f) which is to be expected if participants maintain a representation of the evidence

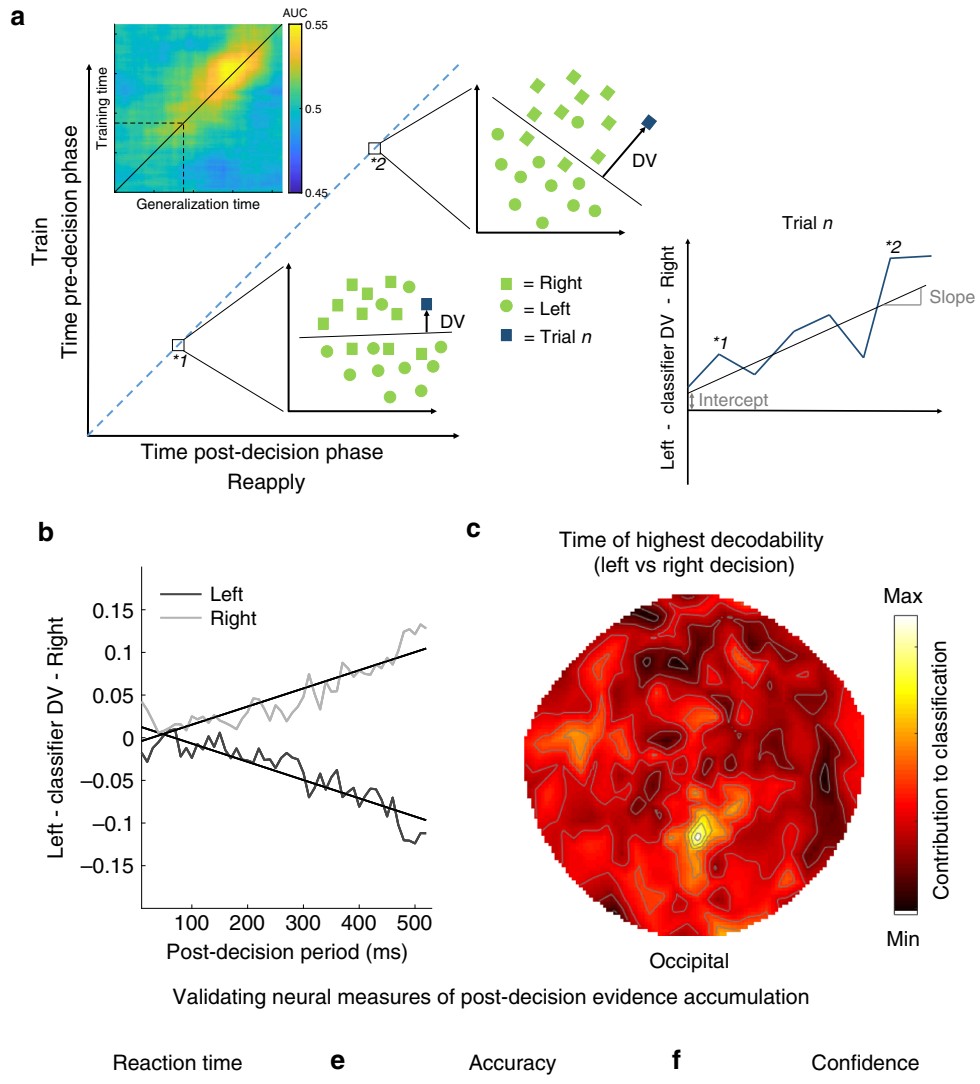

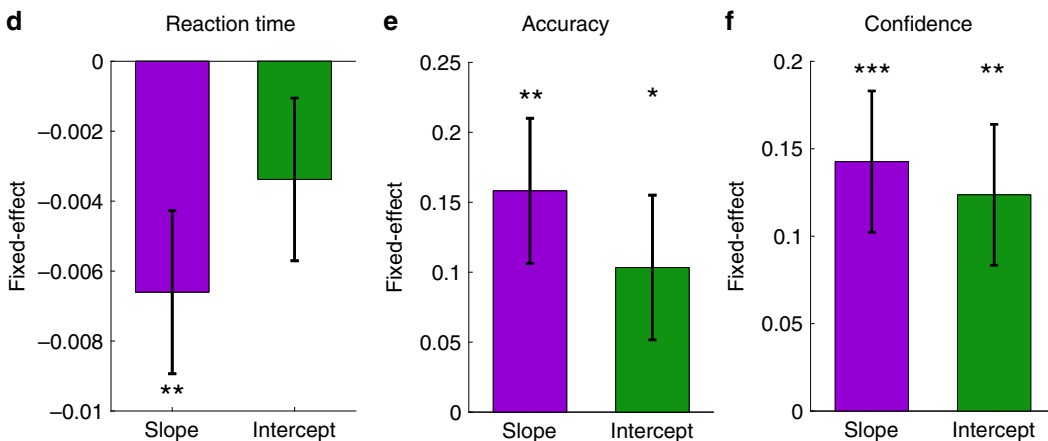

Validating neural measures of post-decision evidence accumulation

obtained in the pre-decision phase, and if the strength of this pre-decisional accumulation predicts the likelihood of being both correct and confident.

We next asked whether specific sensor clusters drive the classifier performance. Previous studies using EEG have identified a centro-parietal event-related potential (the centroparietal positivity or CPP) as a neural marker of internal evidence accumulation[12,30,31]. Accordingly, when identifying the features that contributed most strongly to classifier decoding accuracy (Fig. 3c) we also found that centro-parietal sensors make a

disproportionate contribution to an ability to differentiate between left and right decisions.

Having identified a neural metric of evidence accumulation, we next turned to our central question of whether confidence induces a selective accumulation for choice-consistent information as measured using MEG. As hypothesized, we found that after high confidence (vs. low confidence) decisions, accumulation of neural evidence was facilitated if it was confirmatory, but largely abolished if it was disconfirmatory (Fig. 4a, b). In other words, our MEG analysis reveals that high confidence leads to post-

**Fig. 3 Outline of MEG analysis for quantifying accumulation of post-decision evidence at a neural level (MEG study 3, n = 25 participants). a** We trained a machine-learning classification algorithm on the pre-decision phase using MEG activity to predict left vs. right choices, and reapplied this classifier to the corresponding time point during the post-decision phase. The distance of each trial to the separating hyperplane provides a graded measure of neural evidence for a left or right decision, with changes in the classifier prediction within each trial providing a neural metric of evidence accumulation (see right hand panel). The inset shows the temporal generalization of decoding accuracy from the pre- to post-decision phases, indicating that the pre-decision classifier generalises to the post-decision phase along the major diagonal (i.e. corresponding time-points). AUC = area under the curve, DV = decision variable. **b** Grand average of the left/right classifier prediction in response to post-decision evidence. The light grey line shows the change in neural representation when rightward motion is presented and the black line shows the change in neural representation when leftward motion is presented. Regression lines show fits to the group-averaged data for visualisation purposes. Note that positive classifier values indicate evidence for a rightward decision and negative values evidence for a leftward decision. **c** Contributions of sensors to decoding left versus right decisions. The group average of contributions for each sensor is presented. In line with previous research on the neural correlates of evidence accumulation, sensors in centro-parietal regions made the highest contributions to decodability of (abstract) left versus right decisions. **d–f** Validation of neural metrics of post-decision evidence accumulation. Neural measures of the slope and starting point (intercept) of evidence accumulation extracted from the post-decision phase were entered as simultaneous predictors of (**d**) reaction times (**e**) accuracy and (**f**) confidence of the final decision. Fixed effects from a hierarchical regression model are presented ± SEM. Hierarchical regression (two-tailed): **d** \*\**p* = 0.005; **e** \**p* = 0.045, \*\**p* = 0.002; **f** \*\**p* = 0.002, \*\*\**p* = 0.0004.

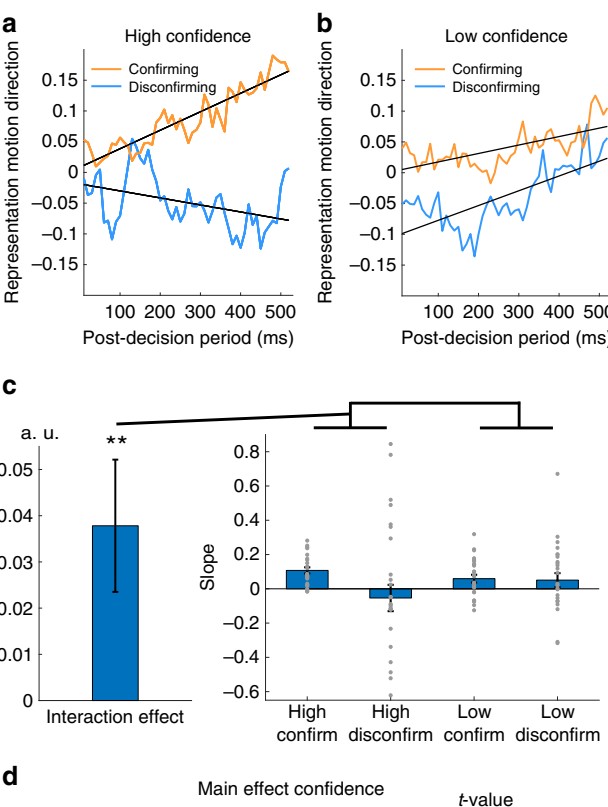

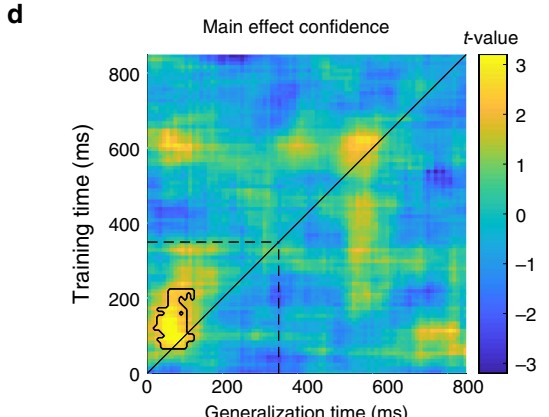

**Fig. 4 MEG analysis investigating the influence of confidence on post-decision evidence processing (MEG study 3, n = 25 participants). a, b** Neural metrics of post-decision accumulation separated into confirming (consistent with initial decision) and disconfirming (inconsistent with initial decision) post-decision evidence and as a function of high (**a**) and low (**b**) initial confidence. More positive values on the y-axis indicate stronger (more veridical) representation of the presented motion. Weighted group averages (grand average) are presented and regression lines are fits to this averaged data. **c** Effects of initial decision and confidence on the slope of neural evidence accumulation in response to post-decision evidence (slope). The righthand panel shows weighted mean values ± SEM for the strength of neural evidence integration (slope) within each condition. Grey dots represent individual participants' data. The lefthand bar shows the fixed effect ± SEM for the initial decision × confidence interaction effect from a hierarchical regression (two-tailed): \*\**p* = 0.008. **d** Effect of confidence on temporal generalization of decoding accuracy from the pre- to the post-decision phase. Higher confidence is associated with higher decodability of the initial decision (i.e. stronger representation of the initial decision, yellow colours). A stronger representation of the initial decision was seen at the beginning of the post-decision period when confidence was high, consistent with confidence shifting a starting point towards the bound of the initial decision. The contoured area represents a cluster of timepoints with a significant main effect of confidence (permutation test, *p* < 0.05 corrected for multiple comparisons). The time window starts with stimulus presentation (0 ms) and ends when the response options are presented (850 ms). Dotted lines indicate the offset of the stimulus (pre- or post-decision stimulus respectively).

decision accumulation becoming "blind" to disconfirmatory evidence. To formally quantify this effect, we entered the slope and starting point of neural evidence accumulation on each trial into hierarchical regression models with initial decision, high vs. low initial confidence and their interaction as predictors. We obtained a significant effect of initial decision ($\beta = 0.042$, $t(8547) = 2.96$, $p = 0.003$) and its interaction with confidence ($\beta = 0.038$, $t(8547) = 2.64$, $p = 0.008$, see Fig. 4c) on slope in the absence of effects on starting point ($p > 0.05$). Consistent with our DDM fits, these results indicate that a confidence-induced confirmation bias is predominantly driven by a selective accumulation of choice-consistent information.

We further reasoned that this approach may remain blind to changes in the starting point of post-decision evidence accumulation because of an asymmetry in evidence availability at the start of the pre- and post-decision phases. In other words, simply reapplying the (non-predictive) classifier weights obtained at the beginning of the pre-decision phase to the same time point in the post-decision phase could render the analysis pipeline blind to

starting point offsets. To address this concern, we evaluated the extent to which the entire timecourse of classifier predictions obtained in the pre-decision phase generalised to the post-decision phase, without making assumptions about their relative timing[32]. This analysis provides insight into how putative processing stages identified in the pre-decision phase are reinstated in the post-decision phase, and crucially how this timecourse is affected by confidence. We found a cluster of time points in which a representation of the initial decision was activated earlier in the post- compared with the pre-decision phase when confidence was high ($p = 0.01$, corrected for multiple comparisons; Fig. 4d). Such early reinstatement of a later processing stage is consistent with confidence enhancing a representation of the initial decision (i.e. shifting a starting point towards the bound of the initial decision) or inducing an expectation for evidence supporting an initial decision at the beginning of the post-decision period. Together these results indicate that confidence changes both the neural representation of evidence for an initial decision at the beginning of the post-decision phase (analogous to a change in starting point) as well as enhancing the processing of evidence supporting an initial decision (analogous to a change in drift rate).

## Discussion
By combining behavioural and neural modelling we provide experimental evidence that holding high confidence in a decision leads to a striking modulation of post-decision processing and the emergence of a behavioural confirmation bias. These findings are consistent with a neural representation of confidence acting as a top-down controller[25] (see Supplementary Note 7 for further analysis) that selectively amplifies processing of choice-consistent information.

A confirmation bias in the current experiment was observed in low-level perceptual decisions with limited emotional or cognitive content, suggesting that choice-induced biases in evidence accumulation represent a core principle of neural information processing[8,33]. In most real-world decisions, additional motivational[34] and social[35] influences (e.g. not revising a decision in order to appear self-consistent) are presumably also in play. These additional influences may amplify, or add to, effects of confidence on post-decisional processing in complex ways. An advantage of starting with an investigation of confirmation biases within lower-level tasks is that the potential for such interactions can be minimized, allowing a focused investigation of the processes that drive post-decisional shifts in evidence accumulation.

Computational modelling of the evidence accumulation process enabled further arbitration between apparently optimal information usage and a confirmation bias, by separating the influence of confidence on post-decisional starting point and drift rate. A shift in starting point is potentially normative as it may reflect the contribution of stronger pre-decision evidence to higher confidence, indicating that participants incorporate both pre- and post-decision evidence when reaching a final decision. In contrast, the influence confidence on drift-rate represents a distortion in the integration of new evidence and thus a classic instance of confirmation bias.

In turn, our usage of MEG recordings in combination with machine learning classification revealed a neural marker of these shifts in post-decision evidence accumulation. This measure complemented our behavioural modelling results and yielded direct support for a hypothesis that confidence alters the way in which the brain accumulates new information, consistent with a selective gating of choice-consistent information.

In the current task, where new evidence is always helpful, this bias against incorporating conflicting post-decision evidence is normatively maladaptive. In other scenarios, however, where new evidence may be distracting and/or actively misleading, a confirmation bias might prove helpful. For instance, previous attempts to explain the value of selective evidence accumulation focused on its role in directing attention towards aspects of the environment with the highest potential for information gain[36,37], or in increasing the robustness of decisions against the influence of noise[26,38]. However, the fact that confidence increases choice-consistent information processing goes against the idea that confirmation bias is itself driven by a need for certainty[3,39]. Instead, we observed the strongest confirmation bias when people were already confident in their decisions.

The study of cognitive biases has remained largely distinct from parallel efforts to understand the processes governing evidence accumulation in simple decisions. We suggest that extending models of evidence accumulation to post-decisional processing enables a unique window onto biases in higher-order cognition[7]. Intriguingly, recent evidence suggests that alterations in post-decision processing are predictive of higher-level attitudes such as beliefs about political issues[13], suggesting that insights gained from the study of confirmation bias in simple decisions can be applied to understand the drivers of polarization and entrenchment across a range of societal issues. For instance, a central role for confidence in shaping the fidelity of evidence accumulation indicates that metacognitive interventions may be one route towards ameliorating this pervasive cognitive bias.

## Methods
**Participants**. Each study contained a different group of participants. We analysed data from 28 participants in study 1 ($M_{age} = 23.8$; $SD_{age} = 6.3$; 16 female) and 23 participants in study 2 ($M_{age} = 25.7$; $SD_{age} = 7$; 12 female). Participants were excluded based on the following set of pre-defined criteria: using the same initial confidence rating more than 90% of time ($N = 3$ in study 1; $N = 2$ in study 2), performance below 55% or above 87.5% correct decisions in one of the pre-decision evidence conditions (see explanation of the experimental conditions below) indicating non-convergence of the staircase procedure ($N = 3$ in study 1; $N = 2$ in study 2).

For the MEG study 3, participants conducted an initial behavioural training session before being screened according to the same criteria reported above. MEG data of a final sample of 25 subjects was analysed ($M_{age} = 24.6$; $SD_{age} = 4.1$; 16 female). Data of four subjects could not be analysed due to technical problems with recording triggers. As we applied machine learning classification algorithms to the neural data in order to decode decisions (left versus right) and confidence (high versus low) it was important that participants showed relatively balanced responses for these two categories. 2 subjects were excluded because they chose one response more than 80% of the time for either the decision or confidence.

In addition to a basic payment (£10 for behaviour and £20 for MEG) participants received a performance-based bonus (up to £5 for behaviour and £8 for MEG). All studies were approved by the Research Ethics Committee of University College London (#1260-003) and all subjects gave written informed consent.

**Stimuli and experimental design**. The psychophysical task was an adaptation of the task used by Fleming and colleagues[18], and programmed in MATLAB 2012a (Mathworks Inc., USA) using Psychtoolbox- 3.0.14. Stimuli were random dot motion kinetograms (RDKs), viewed at a distance of approximately 45 cm. The RDKs were clouds of white dots (0.12° diameter) within a white circular aperture with a radius of 7° on a grey background that lasted for 350 ms. The direction of motion was rightward or leftward along the horizontal meridian. The speed of movement was 5° per second and the density of dots in the whole experiment was set to 60 dots per degree. Each set was replotted three apertures later in which a subset of dots, determined by the percent coherence, was offset from their previous location towards the target movement direction, and another subset was offset in the opposite direction, whereas the rest was replotted randomly.

Unlike in a classical RDK stimulus, dots moved coherently in both the target direction and the opposite direction. The remaining dots moved randomly (percentages described below). We used a psychophysical manipulation of positive evidence to dissociate subjective confidence from objective task performance[27]. In the high positive evidence (HPE) the proportion of dots moving in the incorrect direction was set to 15% and the proportion moving in the correct direction was a higher percentage, staircased to ensure the targeted performance level (see below). In the low positive evidence (LPE) condition the motion coherence of dots moving in the incorrect direction was set to 5%, whereas the dots moving in the correct direction was also staircased to ensure the same performance as in the HPE

condition. The rationale for this manipulation was that accuracy and confidence are usually highly correlated, hindering specific claims about the unique role of confidence. The positive evidence manipulation enabled us to selectively increase confidence while keeping performance constant, thus making it possible to determine a direct effects of changes in confidence on post-decision processing.

All experiments adapted a full 2 (pre-decision positive evidence level) by 2 (post-decision evidence strength) factorial design yielding a total of 4 experimental conditions each corresponding to 90 trials. HPE and LPE stimuli were each followed by one of two post-decision evidence conditions (weak or strong). For the post-decision evidence a constant level of evidence in the incorrect direction was employed (i.e. we did not manipulate the overall amount of positive evidence in the post-decision phase). The post-decision coherence level in the incorrect direction was derived from the averaged staircased pre-decision values as [incorrect coherence LPE + incorrect coherence HPE]/2. Weak post-decision evidence stimuli were created by specifying correct-direction coherence as [staircased correct coherence LPE + staircased correct coherence HPE]/2. Strong post-decision evidence stimuli were then derived by multiplying this coherence level by a factor of 1.3.

**Task procedure**. In every study, participants first performed 180 trials of a calibration phase before performing the main task which consisted of 360 trials (behavioural studies) or 352 trials (MEG study).

In the calibration phase subjects judged whether the dots were moving to the left or to the right side of the screen, without rating their confidence or seeing additional post-decision evidence. The response had to be given within 1.5 s after stimulus offset. LPE and HPE stimuli were randomly interleaved. As described above, the coherence of the target direction was adapted with a staircase procedure to obtain a performance of 60% correct in study 1 and 71% correct in studies 2 and 3[40].

The main task had the same core structure for all studies with slight variations, explained below, to optimize each study for the specific research question and planned analysis. Participants were first presented with a moving dot stimulus before they indicated their initial decision (left or right) together with a confidence rating. In behavioural studies 1 and 2 the decision was indicated by pressing the left or right arrow key on the keyboard and was directly combined with a graded confidence rating (7-point sliding scale between 50% and 100%), where pressing the (same) arrow key again moved a slider along the confidence scale. In the MEG study, subjects first made a left versus right decision, before giving a binary high/ low confidence rating. After this initial decision, participants received a second sample of moving dots (i.e. post-decision evidence) which was always in the same (correct) direction as the pre-decision evidence presentation, but of variable strength. Subjects were instructed that this evidence was bonus information that could be used to inform their final decision and confidence. After the post-decision evidence, participants were again asked to judge the motion direction and indicate their confidence.

**Design alterations in behavioural study 2**. In study 2 we optimized the experimental design to allow drift-diffusion modelling of the second/final decision. While in study 1 subjects had to withhold their final response for 300 ms after the offset of the post-decision evidence (i.e. responding was only possible after this delay), in study 2 participants were able to make their final response freely as soon as they had decided. This allowed us to use response times as a proxy for crossing a decision threshold, which would not have been possible if the response was delayed.

**Design alterations in MEG study 3**. In the MEG study, participants indicated their responses by pressing and up or down button on a keypad with their right thumb. We disentangled the participant's decision (left/right and high/low confidence) from the motor response they had to perform (pressing the up or down key on a key pad), by randomising the mapping between decision options and key presses. Specifically, on any given trial leftward motion could be indicated by pressing the up key and on another trial by pressing the down key. Similarly, high confidence could be indicated in one trial by pressing the up key and in a different trial by pressing the down key. The mapping between decisions and motor responses was revealed once responding was possible, by presenting the letters L or R (and L or L for confidence ratings) above/below the horizontal plane. This approach ensured that decoding of motion direction was not trivially confounded by motor preparation signals. Additionally, we introduced delays of 500 ms after the presentation of each stimulus but before participants were informed about the response mappings to allow decoding analysis to be applied in a time window when subjects could form an abstract decision about motion direction but were not yet able to prepare a response.

**Scoring and bonus payment**. Participants were instructed to rate their confidence as a subjective probability of being correct and were rewarded according to the correspondence between their confidence and task accuracy. An incentive-compatible Quadratic Scoring Rule[41] was applied equally to both the initial and

final decisions:

$$\text{Points} = 100 * \left[ 1 - (\text{correct}_i - \text{confidence}_i)^2 \right] \qquad (4)$$

where $\text{correct}_i$ is equal to 1 on trial i if the choice was correct and 0 otherwise, and $\text{conf}_i$ is the subject's confidence rating on trial i. The Quadratic Scoring Rule is a proper scoring rule in that maximum earnings are obtained by jointly maximizing the accuracy of both choices and confidence ratings. This scoring rule also ensures that confidence is orthogonal to the reward the subject expects to receive for each trial: maximal reward is obtained both when one is maximally confident and right, and minimally confident and wrong. The points gained on each trial were summed and participants were given a £1 bonus payment for every 15,000 points earned. After each block participants were informed of their current total number of points. This was the only performance feedback that was given and subjects did not receive specific information regarding the correctness of their motion direction decisions.

**Multilevel meditation analysis**. A mediation analysis was carried out to examine whether the effect of positive evidence on changes of mind was mediated by a shift in confidence (see Supplementary Notes 5 and 6). We implemented a multilevel mediation model with subjects as random effects, using the Multilevel Mediation and Moderation (M3) Toolbox[42]. Mediation analysis assesses whether covariance between two variables (predictor and dependent variable) is explained by a third mediator variable. Significant mediation is obtained when inclusion of the mediator in the model significantly alters the slope of the predictor-dependent variable relationship (evaluated as the product of the predictor-mediator and mediator-dependent variable path coefficients). In a logistic regression model the two positive evidence conditions (i.e. coded as HPE = 2, LPE = 1) were entered as the predictor variable, changes of mind as the dependent variable (coded as change of mind = 1, no change of mind = 0) with confidence ratings as the mediator variable. We controlled for covariates that potentially could have had a confounding influence on these linkages such as accuracy, reaction time, post-decision evidence strength and the interaction between accuracy × post-decision evidence strength. The following effects of interest were simultaneously tested: the impact of positive evidence on confidence ratings (path a); the impact of confidence ratings on changes of mind, controlling for positive evidence (path b); and the formal mediation of positive evidence on changes of mind by confidence (path a × b). The direct effect of positive evidence on changes of mind before and after controlling for confidence was also estimated (paths c and c', respectively). Parameter estimates for each path (a, b, c, a × b, c') were obtained by bootstrapping 200,000 times with replacement, producing two-tailed p-values and 95% confidence intervals. In a control model in which the predictor and mediator variables were swapped, no mediation effect was found.

**Drift-diffusion modelling**. Drift-diffusion modelling was conducted in Python 2 using Jupyter Notebook (5.50). The model was fit using accuracy coding such that decision boundaries and reaction time distributions corresponded to those for correct and incorrect responses. However, by design, initially correct decisions led to confirming post-decision evidence (because the motion direction was always the same in the pre and post-decision periods) and initially incorrect decisions always led to disconfirming post-decision evidence.

Within the DDM there are two natural ways to account for biases in a decision process: by shifting the starting point towards one of the decision boundaries, or by altering the drift rate to induce a bias in the processing of information. We also considered the possibility that other factors (e.g. decision bound) could be altered, but in initial simulations such changes were unable to explain the observed behavioural patterns. Since it has been reported that confidence might affect boundary separation[29], we included a dependency of the boundary separation on confidence in each of the models (note however that a symmetrical influence on boundary separation cannot explain any choice-dependent effects on changes of mind).

A hierarchical Bayesian variant of the DDM (hDDM) enabled us to investigate the dependencies of the model parameters on the initial decision and confidence on a trial-by-trial basis[43]. The hDDM simultaneously estimates individual parameters drawn from a group distribution using Markov-Chain Monte-Carlo methods. This procedure not only estimates the most likely value of the model parameters but also uncertainty in the estimate. The hDDM toolbox[43] was used to compare 10 hDDMs. The best-fitting model was identified by comparing Deviance Information Criterion scores and ensuring that the wining model adequately fitted the qualitative data patterns (see Supplementary Note 2). A regression analysis was used to investigate the dependency of the starting point and drift-rate parameters on the initial decision (1 = correct decision leading to confirmatory post-decision evidence, −1 = incorrect decision leading to disconfirmatory post-decision evidence), initial confidence (parametrically ranging from −1 to 1) or their interaction.

In all models the drift rate, starting point, non-decision time and boundary separation were fitted hierarchically with individual parameter estimates for each participant, whereas dependencies of starting point and drift-rate on experimental factors were estimated as fixed group-level effects. In all model fits we incorporated an influence of post-decision evidence strength on the drift-rate. First a baseline model was estimated where none of the parameters depended on confidence or an

initial decision. Subsequently, we created three model families that had dependencies of starting point and/or drift-rate on (i) initial confidence, (ii) initial decision or (iii) the interaction of initial confidence × initial decision (i.e. confidence was allowed to amplify or attenuate the influence of the initial decision on the starting point and/or drift-rate). Within each model family we created three different models with dependencies of these variables on starting point, drift-rate or both.

Baseline model (Model 1):

$$\text{Starting point} \sim 1 \tag{5}$$

$$\text{Drift-rate} \sim 1 + \text{post-decision evidence strength} \tag{6}$$

$$\text{Boundary separation} \sim 1 + \text{confidence} \tag{7}$$

Confidence dependency (Model 4):

$$\text{Starting point} \sim 1 + \text{confidence} \tag{8}$$

$$\text{Drift-rate} \sim 1 + \text{post-decision evidence strength} + \text{confidence} \tag{9}$$

$$\text{Boundary separation} \sim 1 + \text{confidence} \tag{10}$$

Initial decision dependency (Model 7):

$$\text{Starting point} \sim 1 + \text{initial decision} \tag{11}$$

$$\text{Drift-rate} \sim 1 + \text{post-decision evidence strength} + \text{initial decision} \tag{12}$$

$$\text{Boundary separation} \sim 1 + \text{confidence} \tag{13}$$

Full model (Model 10):

$$\text{Starting point} \sim 1 + \text{confidence} + \text{initial decision} + \text{confidence} \times \text{initial decision} \tag{14}$$

$$\text{Drift-rate} \sim 1 + \text{post-decision evidence strength} + \text{confidence} \\ + \text{initial decision} + \text{confidence} \times \text{initial decision} \tag{15}$$

$$\text{Boundary separation} \sim 1 + \text{confidence} \tag{16}$$

RTs faster than 200 ms were discarded from the model fits and the outlier probability was set to 0.05, as recommended in previous literature[43,44]. The models were estimated with a Markov chain of 100,000 samples with 50,000 burn-in samples (i.e. discarding the first 50,000 iterations), and a thinning factor of 25, resulting in 2500 posterior samples. To ensure convergence, the posterior traces and their autocorrelation were inspected and the Gelman–Rubin statistic was calculated for each parameter (see Supplementary Table 1). The posterior distributions of the best-fitting model were interrogated to retrieve parameter estimates.

The winning model was characterized by a regression equation that incorporates effects of confidence, the initial decision and their interaction (i.e. the full model) on the starting point and drift-rate. The Deviance Information Criterion scores of all models are shown in Supplementary Fig. 3A. The model parameters of the best-fitting model are shown in Fig. 2d.

**MEG pre-processing**. MEG was recorded continuously at 600 samples/second using a whole-head 273-channel axial gradiometer system (CTF Omega, VSM MedTech), while participants sat upright inside the scanner. Data was segmented into 8200 ms segments from −200 ms to +8000 ms relative to trial onset, where each segment encompassed one trial. Each epoch was aligned to the onset of the trial or, for analysis of the post-decisional phase, was realigned to the onset of post-decision evidence (to minimize any presentation delays that may have occurred during the trial). The data were resampled from 600 to 100 Hz to conserve processing time and improve signal to noise ratio, resulting in data samples spaced every 10 ms. All data were then high-pass filtered at 0.5 Hz to remove slow drift. All analyses were performed directly on the filtered, cleaned MEG signal, consisting of a 273 channel × 821 sample matrix for each trial, in units of femtotesla.

**Generalising a pre-decision classifier to the post-decision phase**. We built a machine-learning classification algorithm to predict participants' decisions on each trial (leftward vs. rightward motion) at each timepoint during the decision phase. Having trained such an algorithm we could then apply it to a distinct set of trials and use the probabilistic prediction of the classifier as a neural DV for leftward versus rightward motion[45,46]. Specifically, we used a support-vector machine (SVM) classifier trained on sensor-level whole-brain activity (normalized amplitude of all MEG channels). The classifier labels were the trial-by-trial choices made by participants (left or right) while the features encompassed a matrix of activity at each MEG sensor ($z$-scored for each time point) at a given time point (average activity over 100 ms window, shifted in steps of 10 ms). The classifier was trained on MEG activity in the pre-decision time phase (e.g. 250 ms after the onset of pre-decision evidence) and then reapplied to the corresponding time point in the post-decision phase (e.g. 250 ms after the onset of post-decision evidence). We

computed the predictions of the classifier across an 850 ms time window, starting with post-decision stimulus onset and ending with the presentation of response options (i.e. when the mapping between choices and motor responses was revealed).

We used linear kernels and a default regularization parameter of $C = 1$ within the svmtrain/svmpredict routines of libsvm[47]. A leave-one-out procedure was used, training the classifier on all trials except one (using pre-decision data only) and testing it on the left-out trial (using post-decision data). Training the SVM results in a hyperplane that best separates the two classes of trials (see Fig. 3a) in a high-dimensional space. If a trial is far away from this hyperplane it is unlikely to be a misclassification, while trials that are close to the hyperplane might easily be misclassified. Thus, the distance to the hyperplane represents the decodable evidence for a decision and can thus be used as a graded measure of the neural DV[45,46].

After reapplying the classifier to every trial and time point during the post-decision phase, we obtained a timeseries of neural evidence accumulation within each trial (see Fig. 3a, right panel). We focussed on the time from the onset of the post-decision stimulus to the timepoint of peak decodability at which the pre-decision classifier best generalized to the post-decision phase. The accumulation process can be summarized by fitting a linear regression to the time series (see Fig. 3a, right panel) on each trial, where the slope is analogous to the drift rate in a DDM, and the intercept analogous to the starting point. A positive slope corresponds to a change of the neural DV towards predicting rightward motion decisions while a negative slope corresponds to a change towards leftward motion (see Fig. 3b). By taking the absolute value of these slope values (i.e. reversing the sign on trials in which leftward motion was presented), we could derive a general index for the sensitivity of the neural DV to the motion direction presented on the screen (see Fig. 4a, b).

Based on our behavioural findings we expected that both the slope and the intercept would be influenced by the interaction of initial decision (confirmatory post-decision evidence = 1; disconfirmatory post-decision evidence = −1) × confidence (low confidence = −1; high confidence = 1). Thus, we entered the initial decision, confidence and their interaction as simultaneous predictors in a hierarchical regression model.

**MEG topography contributing to classification accuracy**. To explore which brain areas carried the information about evidence for a left versus a right decision (or high versus low confidence as reported in the Supplementary Note 6), we trained a SVM classifier for each participant at the time point of highest decodability (see Supplementary Fig. 6 for the whole timeline) using subsets of 30 randomly selected sensors and repeated this procedure 2500 times. The contribution of each sensor $s$ was taken to be the mean of all prediction accuracies achieved using an ensemble of 30 sensors that included $s$[48,49].

**MEG temporal generalization**. The extent to which a classifier trained on neural data obtained from one time point generalizes to other time points can provide insight how mental representations change over time[32]. We utilized this temporal generalization method to formally test whether the same processing steps (leading up to a decision) occur at similar times in the pre- and post-decision phases (see Supplementary Fig. 9 for temporal generalization restricted to the pre-decision phase). Most critically, we also investigated whether this processing cascade was altered by participants' confidence in their choice.

For the temporal generalization analysis we trained our classifier on every timepoint in the pre-decision phase and tested it on every timepoint in the post-decision phase yielding a 2D matrix of decoding accuracy (see Fig. 3a top-left panel). A fourfold stratified cross-validation was implemented for each subject and repeated 100 times to account for potential random biases in assigning trials to folds. Through this stratification we obtained a balanced number of trials within each condition in each fold (left/right decision, high/low confidence, change/no change of mind, and all combinations of these factors). Classifiers were trained on three out of four folds and tested on the left-out fold. Decoding accuracy was determined by the area under a Receiver Operator Curve (AUC) that sought to predict the decision based on the continuous DV outputted by the classifier. Decoding accuracy was calculated separately for the four different conditions (low confidence and change of mind; high confidence and change of mind; low confidence and no change of mind; high confidence and no change of mind). Importantly, classification accuracy was based on how well the initial decision (rather than the final decision) could be predicted based on neural data. Since we are dealing with a two-class decoding problem one can directly infer the decoding of the alternative decision from the classification accuracy of the initial decision.

We estimated the main effect of confidence on decoding accuracy to isolate confidence-induced changes in temporal generalisation from the pre- to post-decision phase. We used a cluster-based permutation test[50,51] to determine statistical significance ($p < 0.05$, corrected for multiple comparisons). We calculated the contrast of high > low confidence averaging over change/no change of mind trials [[high confidence and no change of mind − low confidence and no change of mind] + [high confidence and change of mind − low confidence and change of mind]]. We identified adjacent timepoints all individually exceeding $t$-values corresponding to $p < 0.05$ uncorrected, and stored the sum of $t$-values for each cluster. We then applied a sign-flip permutation test (randomly switching the

contrast direction for a subset of subjects of the sample, i.e. low-high instead of high-low) and repeated this procedure 1000 times. The distribution of summed $t$-values over all permutations built the null distribution for our statistical test. If the observed sum of $t$-values within a cluster exceeded the 5% quantile of this distribution (separately calculated for negative and positive values) we labelled this cluster as showing a significant main effect of confidence in this portion of the temporal generalisation matrix.

**Reporting summary**. Further information on research design is available in the Nature Research Reporting Summary linked to this article.

## Data availability
Anonymised data and code are available at a dedicated Github repository [https://github.com/MaxRollwage/NatureCommunications]. The source data underlying Figs. 1b–d, 2b–d, 3b, d–f, and 4a–d are provided as part of this repository. A reporting summary for this Article is available as a Supplementary Information file.

## Code availability
Code supporting this study are available at a dedicated Github repository [https://github.com/MaxRollwage/NatureCommunications].

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

## Acknowledgements

We thank Dr Sam Ereira for help with implementing the machine learning MEG analysis. M.R. is a predoctoral Fellow of the International Max Planck Research School on Computational Methods in Psychiatry and Ageing Research. The participating institutions are the Max Planck Institute for Human Development and University College London (UCL). The Wellcome Centre for Human Neuroimaging is supported by core funding from the Wellcome Trust (203147/Z/16/Z). S.M.F. is supported by a Sir Henry Dale Fellowship jointly funded by the Wellcome Trust and the Royal Society (206648/Z/17/Z). T.U.H. is supported by a Wellcome/Royal Society Sir Henry Dale Fellowship (211155/Z/18/Z), a grant from the Jacobs Foundation (2017-1261-04), the Medical Research Foundation, and a 2018 NARSAD Young Investigator grant (27023) from the Brain & Behaviour Research Foundation.

## Author contributions

S.M.F. and M.R. conceptualized the study. M.R. developed the methodology under supervision of S.M.F. and with input from T.U.H. A.L. and M.R. conducted the experiments. M.R. analysed the data under supervision of S.M.F. and with input from R.M. and T.U.H. M.R. and S.M.F. wrote the first draft of the manuscript which was revised and edited by A.L., T.U.H., R.M. and R.J.D.

## Competing interests

All authors declare no competing interests.
