## [Peer Review File · Nature Communications]

Editorial Note: Parts of this peer review file have been redacted as indicated to remove third-party material where no permission to publish could be obtained.

Reviewers' comments:

Reviewer #1 (Remarks to the Author):

In this paper Rollwage et al examine how participants update their confidence in perceptual choices in light of sensory evidence received after the initial decision has been made. A combination of behavioural analyses, computational modelling and MEG signal analysis point to participants engaging in a form of confirmation bias whereby evidence consistent with a preceding choice is selectively amplified alongside strategic adjustment of the decision bounds.

This is a beautifully designed experiment matched by a beautifully written manuscript. The results are certainly of interest to the community and the authors can be applauded for applying such a principled, mechanistic framework to the study of metacognition. I do have a number of substantial concerns regarding how the authors pitch the experiment as well as some queries regarding the modelling and the meaning of the neural data that I hope the authors can address.

1. In the abstract in particular the authors implicitly make the very strong claim that the kind of confirmation bias highlighted in their perceptual discrimination task can be considered analogous to that associated with climate change denial. In fact, the authors segue directly from mentioning climate change to stating that their experiment will 'identify the drivers of a confirmation bias' and 'show that holding high confidence in A DECISION...'. There is nothing here to flag for the reader that we have switched from thinking about highly complex high-level decisions to elemental perceptual decisions. Similarly, beyond asserting the link and citing the authors recent Current Biology paper, there is little in the Introduction to reassure/motivate the reader to believe that studying biases in perceptual decisions can shed light on real-world confirmation bias. In fact, the authors note that an advantage of perceptual decision tasks is that they strip away emotional and cognitive influences. This is true and indeed advantageous in many respects but in the present context it is also troubling because those emotional and cognitive influences might, for all we know, be the principal determinants of confirmation bias when it comes to issues like climate change. I think the authors do a good job of demonstrating a form of perceptual confirmation bias that is interesting and important in its own right but I fear that they are overextending themselves in trying to sell the general implications. The drift bias observed here could well be particular to perceptual decisions or, at best, might reflect one potential contributor to confirmation bias.

2. It would be nice to hear the authors' regarding the degree to which the present findings would extend beyond the particular RDM task variant implemented here. The fact that subsets of dots moved coherently in both directions may have encouraged/forced participants to selectively attend to one or other motion channel in order to avoid making incorrect responses. This could be the principal driver of the drift bias. Ideally the authors would provide some confirmation of their results using a different

paradigm e.g. standard variant of the dot motion task with just one direction of coherent motion but I am open to being convinced otherwise

3. In modelling behaviour it is my understanding that the authors effectively assume that participants initiate a second decision process during the post-decision window and this process is subject to bias in the form of starting point and drift rate adjustments. Did the authors consider other formulations? For example, Pleskac and Busemeyer (2010) developed a DDM model variant that could account for final confidence reports by assuming that a single evidence accumulation process determines both the first- and second-order reports. The influence of such a process on second-order RT and accuracy can almost certainly be captured by the present authors' approach but their approach misses the opportunity to account for the final confidence reports. I don't see this as a major flaw but it would be interesting to hear the authors' thoughts on this point

4. I think it would be important for the authors to provide further discussion of the meaning of the neural signals. No topographies are provided to give any sense of where in the brain such signals might emanate or what class of signal they represent. The authors allude to the possibility that they may reflect an abstract evidence accumulation process but they are not at all clear on this point. For example, if the signals do reflect an abstract accumulation process then why do they evolve in opposite directions for left vs right choices? The authors seem to do a good job of excluding motor preparation signals through their task design. One additional control analysis that would be useful here would be to analyze premotor beta-band activity (see work of Donner et al e.g. 2009 Current Biology) and verify that it does not discriminate between the choices because a potential concern could be that participants prepare a provisional action in advance of the response cue. Can the authors rule out that these signals might reflect sensory activity (e.g. MT) as opposed to decision formation?

Reviewer #2 (Remarks to the Author):

The work explores the role of decision confidence as a potential driver of confirmation bias. More specifically, the authors investigate how confidence following a primary decision might affect the process of evidence accumulation on a subsequent (secondary) decision. They show that following highly confident choices the integration of confirmatory evidence in the secondary decision is amplified, while disconfirmatory evidence processing is minimized.

The topic is interesting and the paper could potentially add to recent evidence demonstrating how sequential trial dependencies (including confirmation biases) might be the product of confidence-weighted accumulation of relevant evidence (e.g. Urai et al., eLife 2019, Talluri et al., Curr Biol 2018). I do, however, have a number of conceptual and methodological concerns that the authors should consider (see below).

As a general remark, I found it challenging to follow the paper given the current (heavily condensed) format, presumably reminiscent of an earlier submission to a different journal. The paper can benefit

from a proper introduction to better motivate the work and the proposed hypothesis. Similarly, results should be presented (and discussed) in sufficient detail to allow the reader to fully digest the main contributions of the work (I would also move some of the supplementary figures in the main text to facilitate this).

In the same spirit, excluding subjects that did not exhibit a “positive evidence” manipulation on changes of mind is misleading, since it conceals inter-individual differences and inflates the generality of the presented findings (there was in fact no statistically significant effect of the manipulation when all subjects were included; supplementary figure 2). It might be more appropriate to openly present the full extent of the data in the main text.

The proposed modelling framework appears to suggest that even following low confidence trials, subjects are more likely to move their starting points (and/or drift speed) towards the initially chosen option (as in illustration of Fig. 2A) – though to a lesser extent than following more confident choices. This is a bit counterintuitive. One might expect that confirmation bias is mainly present for confident trials, while following less confident choices, starting point/drift speed might be more neutral (or even biased in the opposite direction). Have the authors explicitly tested for this possibility? Though I understood the broader model family definitions, I’m still unsure (after reading the methods and supplement) what the exact properties of the 10 different model parameterizations were.

In this spirit, I believe it would be useful to further divide up RTs based on the accuracy of the initial choice (as was done in Fig. 2B for accuracy). For example, in the alternative formulation proposed above, when subjects are highly unsure in their primary decision, the relative speed with which they repeat or reverse their choice in the secondary decision could be different than the current model’s prediction.

Following from this, in addition to the formal model comparisons it would be important to quantify the goodness of fit of the winning model (R^2 or similar). In other words, the winning model might still be a somewhat poor fit to the data. Visualizing the average RT fits as in Fig. 2C, masks the extent to which the model captures the full profile of the RT distributions (skewness, heavy tails etc). It would be more informative if the full RT distributions from the data and model simulations were superimposed (separately for each of the relevant conditions) to allow a comprehensive comparison.

Regarding the MEG analysis, I’m a bit surprised that one could discriminate the choice (left vs right). This implies that there exist separate accumulators for left and rightward choices in *spatially* distinct regions such that the MEG can resolve them – this seems unlikely based on the architecture of known decision accumulator regions. The other possibility is that the classifier exploits the encoding of left/right evidence in sensory cortex, which also seems unlikely given that different motion directions are encoded within the same area(s) (MT/MST) and MEG would not be able to resolve them. One thing that is unclear from the methods is whether subjects used the same hand to indicate their choice or whether different choices were mapped onto separate hands, which can have implications on how

these signals are interpreted.

One thing that is missing from the paper and could potentially resolve some of these issues are the scalp topographies associated with the signal extracted from the classifier. Since the work uses linear SVMs it would be trivial to estimate a forward model to see how the extracted signal projects back onto the sensors. The process of evidence accumulation has a very prototypical scalp topography as shown in a number of papers from several groups (centroparietal positivity). This could further validate whether the observed signal is indeed a reliable metric for the process of evidence accumulation.

Another observation that casts some doubt on the interpretation of the proposed accumulator signals is the negative going slopes on some conditions. Why would this signal “ramp down”? Unlike single neuron response profiles (as in LIP for example) coding for specific choices, MEG signals reflecting populations responses over groups of neurons are unlikely to exhibit such selectivity. This point is also tightly linked to my earlier comment regarding the origin and nature of the discriminating activity.

On a more methodological note, the approach for estimating slopes assumes an one-to-one mapping between pre- and post-decision periods, but the way decision dynamics unfold (e.g. onset of accumulation) might be different across the two periods. For example, the hot spot above main diagonal in fig 3A, suggests the generalization in the post-decision period happens earlier. An alternative approach would be to identify individual peak generalization performance, even if the relevant time point appears off the main diagonal, and work backwards along that diagonal (i.e. follow a new trajectory).

There seems to be a discrepancy between the model and the MEG predictions on starting point? Is this an issue with the modeling or the extraction of the relevant MEG signals? Please note that starting point manipulations (e.g. changes in prior beliefs might manifest more in the frequency content of the signal rather than the evoked responses on which the classifier was trained on).

If understood the analysis correctly, Fig. 3A shows results whereby training was done on pre-decision data and the classification weights were used to predict the secondary choice on post-decision data, whereas in Fig. 3E the same procedure was adopted to predict the *initial* choice on post-decision data. Assuming that's true, do the effects from 3E predict the slopes in the analysis presented in 3A? In other words, the effects presented in 3E can be seen as a proxy of the initial influence of confidence on the secondary decision.

Are the 2D images shown in fig. 3A, E from individual subjects or the population? For Fig. 3E, which of the four trial types does this panel correspond to? What happens to the other conditions? For example, are there any interesting dissociations that can further validate the proposed effects?

Minor:

I would tone down the examples given in the abstract (and at the beginning of the paper) regarding the

utility of the observed effects in complex societal issues. The task used here involves a simple perceptual discrimination task and it remains unclear whether the reported effects (and underlying mechanisms) extend to other types of decisions involving great emotional and cognitive influences.

Reviewer #3 (Remarks to the Author):

In this work Rollwage and colleagues report the result of three studies showing that high confidence in a first decision induces a confirmation bias in a second (linked) decision. Using drift diffusion modeling they demonstrate that confidence affects the starting point and drift rate of the second decision. Using MEG recordings, they show that after high confidence decisions choice inconsistent evidence is largely ignored. I think this is an interesting piece of work, with also quite elaborated and consistent additional analyses reported in the supplements. Below, I outline a couple of reservations I had, mostly related to the strength of some claims, fitting procedures and overall clarity.

Major comments

1. The effect of confidence on the drift rate in Experiment 2 was confusing. Traditionally, the DDM is represented with bounds for left versus right responses. With such implementation, a confirmation bias would be expressed as a stimulus-independent (but previous choice dependent) constant added to the mean drift (“a drift criterion”). However, the authors instead model the two bounds as chosen versus unchosen, so that the confirmation bias boils down to a simple drift rate effect. I think explaining this more carefully would help to avoid quite some confusion. In the methods, it is stated that accuracy coding was used. However, if initially the wrong option is selected with high confidence, a higher drift rate implies a higher probability of ending up at the correct bound (whereas the data shows less changes of mind with high confidence).
2. I am a bit worried about the fitting procedure of the DDM. The posterior distributions of the DDM fits shown in 2D look quite noisy, and are based on a rather low number of posterior samples (1000, which is only a fraction of what is done usually). The very large burn-in and thinning seem to suggest difficulties with the fitting procedure. It would be good if the authors could demonstrate model convergence formally (i.e., by computing the Gelman-Rubin statistic), show the autocorrelation patterns, and provide any other relevant information that can help to alleviate any concerns there.
3. The authors make a very strong claim about showing a “causal” effect of confidence on post-decision processing. The findings are certainly in line with an explanation in such terms, but I think a bit more caution is warranted, and some claims about causality should be toned down a bit. For example, the effect of confidence in Figure S6B might be due to any other variable jointly influencing confidence and changes of mind.
4. Although I thought the MEG analyses very elegantly demonstrated a neural confirmation bias, they were a bit of a “black-box”. We know quite a lot about the neural implementation of perceptual decision

making (e.g., CPP, motor beta), and not so much yet about neural implementation of confidence. It would be informative to provide some more insight about what the decoder is picking up on. For example, scalp distributions might be useful to see whether the choice decoder is “just” picking up on something like the CPP or motor beta, or whether something else is going on. The same goes for the decoding of confidence.

Minor comments

5. For some temporal generalization matrices, it is unclear whether stats are missing or whether there are no significant clusters (e.g., Figure S8A and S9). This should be clear.

6. Because much of the introduction focused on evidence accumulation models, it was confusing that Figure 1 displays a design which –on the look of it- does not feature two-choice responses nor speeded response, both of which are essential for fitting the DDM. It would be good if the information that it is in fact a 2CRT task is not hidden in the methods.

7. I might have missed it, but did Experiments 2 and 3 also show an increase in confidence and a decrease in changes of mind for HPE? In the main text, only significant mediation in all three experiments is mentioned.

8. The colors in Figure 2 were ambiguous. Why are different colors used to in panel A to demonstrate the effect of starting point versus drift rate? The same colors appear in panel B without any relation, and again in panel C but now the mapping is even reversed compared to A.

Detailed reply to the reviewers' comments

Reviewers' comments:

Reviewer #1 (Remarks to the Author):

In this paper Rollwage et al examine how participants update their confidence in perceptual choices in light of sensory evidence received after the initial decision has been made. A combination of behavioural analyses, computational modelling and MEG signal analysis point to participants engaging in a form of confirmation bias whereby evidence consistent with a preceding choice is selectively amplified alongside strategic adjustment of the decision bounds.

This is a beautifully designed experiment matched by a beautifully written manuscript. The results are certainly of interest to the community and the authors can be applauded for applying such a principled, mechanistic framework to the study of metacognition. I do have a number of substantial concerns regarding how the authors pitch the experiment as well as some queries regarding the modelling and the meaning of the neural data that I hope the authors can address.

We thank the Reviewer for this positive evaluation and are grateful for their detailed suggestions that we now address in full.

1. In the abstract in particular the authors implicitly make the very strong claim that the kind of confirmation bias highlighted in their perceptual discrimination task can be considered analogous to that associated with climate change denial. In fact, the authors segue directly from mentioning climate change to stating that their experiment will 'identify the drivers of a confirmation bias' and 'show that holding high confidence in A DECISION...'. There is nothing here to flag for the reader that we have switched from thinking about highly complex high-level decisions to elemental perceptual decisions. Similarly, beyond asserting the link and citing the authors recent Current Biology paper, there is little in the Introduction to reassure/motivate the reader to believe that studying biases in perceptual decisions can shed light on real-world confirmation bias. In fact, the authors note that an advantage of perceptual decision tasks is that they strip away emotional and cognitive influences. This is true and indeed advantageous in many respects but in the present context it is also troubling because those emotional and cognitive influences might, for all we know, be the principal determinants of confirmation bias when it comes to issues like climate change. I think the authors do a good job of demonstrating a form of perceptual confirmation bias that is interesting and important in its own right but I fear that they are overextending themselves in trying to sell the general implications. The drift bias observed here could well be particular to perceptual decisions or, at best, might reflect one potential contributor to confirmation bias.

Point 1

We agree and thank the Reviewer for this suggestion to temper this aspect of the manuscript. We have removed reference to climate change in the abstract and adapted multiple sections of the revised manuscript to make clear the aspect of confirmation bias that our task allows us to focus on. We now state limitations of our results, noting other factors are likely to contribute to real-world confirmation biases.

The following changes were implemented:

We have removed reference to climate change and deleted the following sentence from the abstract (page 2):

“... proving a pathway towards understanding and counteracting the drivers of intransigent and entrenched beliefs across a range of societal issues.”

We now include a more extensive introduction where we make explicit a potential link between confirmation bias in higher order beliefs and perceptual tasks. Crucially, we also include greater justification for using a perceptual task to study the computational and neural mechanisms underlying confirmation bias (page 3):

“The philosopher Bertrand Russell opined “The most savage controversies are about matters as to which there is no good evidence either way”. While this view applies in some situations, even more troubling are instances where polarization and entrenchment of opinion persists in the face of contrary evidence, exemplified by debates on climate change and vaccinations. This polarization is most evident when opposing parties are highly confident in their positions^{1,2}. A psychological-level explanation for this entrenchment is the idea that people selectively incorporate evidence in line with their beliefs, known as confirmation bias³. Although an extensive literature has documented this bias in behaviour^{3,4}, the underlying cognitive, computational and neuronal mechanisms are not understood.

So far, an investigation of confirmation bias has been restricted largely to scenarios involving complex real-world beliefs such as political attitudes⁴⁻⁶. However, the complexity of such higher-order beliefs makes it difficult to disentangle the various contributors to biased information processing. For instance, people may have a strong personal investment in their political opinions, leading to a significant motivation to discount new information that goes against their beliefs. Intriguingly, confirmation biases have recently been demonstrated in low-level perceptual tasks⁷⁻⁹ that are unlikely to evoke such motivated reasoning. These studies indicate a source of confirmation bias may be a generic shift in the way the brain incorporates new information. Here we adopt such a task to study the computational and neural basis of post-decisional shifts in sensitivity to choice-consistent information.

Perceptual decision-making is well-described using sequential sampling models which assume the brain accumulates noisy evidence for each choice option to a decision bound¹⁰. This accumulation process is thought to be supported by neuronal populations in parietal and prefrontal cortex^{11,12}. Importantly, while perceptual tasks allow tight control over the processes involved, they also permit generalisation to more complex decisions¹³⁻¹⁵, and similar principles appear to underlie choice and confidence formation in both simple and more complex tasks^{16,17}. However while the processes underlying perceptual decision-making have been studied in detail, little is known about the mechanisms governing accumulation of evidence after a choice has been made, or how such processing is shaped by pre-existing beliefs and confidence^{7,17-23}.”

Finally, we now include additional discussion on motivational or societal drivers of confirmation bias and discuss how our effects might be related to these additional determinants in real-world settings:

“In most real-world decisions, additional motivational^{13,4} and social¹⁵ influences (e.g. not revising a decision in order to appear self-consistent) are presumably also in play. Crucially, these additional influences are likely to amplify, or add to, an effect of confidence on post-decisional processing rather than reduce it. Indeed, it is plausible that alterations in post-decisional processing interact with other influences on confirmation bias in complex ways. An advantage of starting with an investigation of confirmation biases within lower-level tasks is that the potential for such interactions can be minimized, allowing a focused investigation of the processes that drive post-decisional shifts in evidence accumulation.”

2. It would be nice to hear the authors' regarding the degree to which the present findings would extend beyond the particular RDM task variant implemented here. The fact that subsets of dots moved coherently in both directions may have encouraged/forced participants to selectively attend to one or other motion channel in order to avoid making incorrect responses. This could be the principal driver of the drift bias. Ideally the authors would provide some confirmation of their results using a different paradigm e.g. standard variant of the dot motion task with just one direction of coherent motion but I am open to being convinced otherwise

Point 2

We thank the Reviewer for this astute comment on our experimental paradigm and the potential confound. Indeed, in our paradigm subsets of dots moved coherently in both directions and in principle this choice of stimulus could influence the generality of our results. However, there are several reasons why we think that our results will also hold true in the standard version of the RDM, or even in tasks using different perceptual stimuli.

First, our results build closely on previous literature in perceptual decision-making that used somewhat different setups, and yet still observe a similar confirmation bias. For instance, Talluri et al. (2018) showed that a binary decision in a standard RDM task induced a selective gain for choice consistent post-decision evidence. In this sense their findings are in strong alignment with our results and indicate that selective amplification of choice-consistent evidence is also observed when using a more standard unidirectional RDM setup. Moreover, they also report a similar effect in a numerical averaging task, lending support to the generality of the effect. In this respect, we believe that previous literature has reassuringly shown that a selective increase in processing of confirming information (change in drift-rate) is a crucial driver of a perceptual confirmation bias independent of the exactly used task.

We stress however, that the main contribution of our studies is not only to show that a perceptual confirmation bias manifests in a selective amplification of confirming post-decision evidence, but a demonstration that this effect is modulated (enhanced) by the confidence people hold in their decisions. Moreover, we reveal a neural mechanism underlying this behavioural phenomenon and thus extend on findings in previous literature.

3. In modelling behaviour it is my understanding that the authors effectively assume that participants initiate a second decision process during the post-decision window and this process is subject to bias in the form of starting point and drift rate adjustments. Did the authors consider other formulations? For example, Pleskac and Busemeyer (2010) developed a DDM model variant that could account for final confidence reports by assuming that a single evidence accumulation process determines both the first- and second-order reports. The influence of such a process on second-order RT and accuracy can almost certainly be captured by the present authors' approach but their approach misses the opportunity to account for the final confidence reports. I don't see this as a major flaw but it would be interesting to hear the authors' thoughts on this point

Point 3

The Reviewer's comment has prompted further reflection. Our formulation of the drift-diffusion model in fact shares considerable conceptual overlap with previous modelling attempts of post-decision evidence accumulation and its links to confidence judgments (for instance Pleskac & Busemeyer, 2010 and van den Berg et al., 2016). In keeping with these formulations, our modelling approach is consistent with a view that (partly) overlapping information contributes to both choice and confidence. For instance, trials in which a lot of evidence is accumulated will lead to high (initial) confidence.

In Pleskac and Busemeyer's (2010) model a post-decision period was defined as the time between a decision and provision of a confidence rating. In this model (and the tasks that were developed to test model predictions such as those reported in van den Berg et al. 2016), no additional evidence is presented after a decision and only evidence which is still in a "pipeline" continues to accumulate during this period. In our study there is a clear break between the pre- and the post-decision periods, and we present new exogenous evidence following a subject's initial decision. Therefore, our task is qualitatively different to the situation modelled by Pleskac and Busemeyer. However, in keeping with their approach, we assume also that a second decision process is initiated during the post-decision period and that this accumulation process is strongly coupled to a pre-decisional accumulation process.

Rather than assuming such coupling a priori, we adopted a flexible modelling approach that does not rely on an assumption that the post-decision accumulation process starts off where the accumulation ended during the pre-decision phase. While this may occur in practice, there are reasons to doubt such a continuation of accumulation – for instance, LIP recordings have shown that firing rates are reset once a decision bound is hit (e.g. Roitman & Shadlen, 2002 J Neuroscience), in keeping with absorbing bounds in the DDM. Instead, links between pre- and post-decision accumulation can be flexibly captured by a dependency of the post-decision starting point on the initial decision, as well as an interaction between initial decision \times initial confidence. We now make this link more explicit in the manuscript and explain our modelling approach in more detail (page 6):

“We considered two potential mechanisms through which confidence might reduce changes of mind. First, confidence might reflect a shift of the starting point of post-decision accumulation closer to the bound associated with an initial decision, consistent with a continuation of pre-decisional evidence accumulation (influence on starting point; Figure 2A upper panel). Second, confidence may induce selective accumulation of evidence in line with an initial decision (influence on drift rate; Figure 2A lower panel) – a clear instance of confirmation bias.”

We agree with the Reviewer that it is interesting to also model the distribution of final confidence ratings following post-decisional processing. In our study we were interested primarily in the effects of (mid-trial) confidence on post-decisional accumulation, so we would prefer to defer this question for future study.

4. I think it would be important for the authors to provide further discussion of the meaning of the neural signals. No topographies are provided to give any sense of where in the brain such signals might emanate or what class of signal they represent. The authors allude to the possibility that they may reflect an abstract evidence accumulation process but they are not at all clear on this point. For example, if the signals do reflect an abstract accumulation process then why do they evolve in opposite directions for left vs right choices? The authors seem to do a good job of excluding motor preparation signals through their task design. One additional control analysis that would be useful here would be to analyze premotor beta-band activity (see work of Donner et al e.g. 2009 Current Biology) and verify that it does not discriminate between the choices because a potential concern could be that participants prepare a provisional action in advance of the response cue. Can the authors rule out that these signals might reflect sensory activity (e.g. MT) as opposed to decision formation?

Point 4

The reviewer makes a very helpful suggestion. As a result, we have made a series of changes to the manuscript including carrying out a new analysis to probe the nature of neural signals tracking post-decision evidence, as follows:

1. The reason that the classification predictions evolve in different directions for left versus right choices is a result of how the classification algorithm is set up – positive values predict a rightward decision and negative values predict a leftward decision. We now explain this in more detail (page 10):

“We then applied the trained classifier to brain activity at the corresponding time point in the post-decision time window, enabling us to derive a probabilistic prediction of internal evidence favouring a leftward versus rightward decision (see Figure 3A left panel), where the strength of this prediction indexes neural evidence in favour for the respective decision. Positive values indicate prediction of a rightward decision and negative values indicate prediction of a leftward decision (see Figure 3B). We next fitted a linear regression to the time series of classifier predictions within each trial (see Figure 3A right panel) to obtain a trial-by-trial neural measure of the starting point

(intercept) and drift rate (slope). These measures of evidence accumulation (slope) should be highly responsive to the presented motion direction during the post-decision period, and we show this was indeed the case (hierarchical regression: $\beta = .08$, $p < 10^{-14}$, Figure 3B).

The slopes extracted from this analysis are signed, such that positive values indicate stronger prediction of a rightward choice and negative values stronger evidence for a leftward choice. In order to obtain an unsigned metric of evidence accumulation strength, we flipped the sign of the slopes extracted from trials in which leftward motion was presented (we conducted the same flip for the intercept to obtain an unsigned metric of the starting point). This measure quantifies a propensity to correctly integrate the presented information, by tracking an internal decision variable (analogous to a drift rate in an accuracy coded drift-diffusion model as used in Figure 2), where higher values indicate stronger sensitivity to the presented stimulus.”

2. We have now moved parts of the Supplementary Information into the main manuscript (see new Figure 3) where we report additional analysis to validate that our neural measure of evidence accumulation fulfils benchmark criteria associated with an internal evidence accumulation signal (page 11):

“A neural analogue of the drift-rate (or change in internal decision variable) should be related to characteristic features of the observer’s decision. Specifically, stronger internal evidence accumulation should be related to a higher likelihood of having made a correct decision¹², faster response times¹⁰ and higher confidence¹¹. In order to check whether our classifier predictions satisfied these criteria for metrics of internal evidence accumulation, we entered both the slope and intercept of the post-decision accumulation process as simultaneous predictors in a hierarchical regression model to predict a) choice accuracy, b) reaction times and c) confidence of the final decision (see Supplementary information for similar analysis of the pre-decision period). A steeper slope predicted faster reaction times ($\beta = -0.007$, $p = .004$, see Figure 3C), a higher likelihood of a correct decision ($\beta = .16$, $p = .002$, see Figure 3D) and higher confidence ($\beta = .14$, $p = .0004$, see Figure 3F) in the final decision. We also observed significant effects of the intercept on accuracy ($\beta = .1$, $p = .045$, see Figure 3D) and confidence ($\beta = .12$, $p = .002$, see Figure 3F) which is to be expected if participants maintain a representation of evidence obtained in the pre-decision phase, and if the strength of this pre-decisional accumulation predicts the likelihood of being both correct and confident.”

3. As suggested by this reviewer, we conducted additional analysis to explore the topography of classifier predictions (Figure 3C and Supplementary Figure S6C-D). In line with previous literature identifying a centro-parietal positivity (CPP; e.g. O’Connell et al. 2012) as a neural marker of abstract evidence accumulation, we find that activity in centro-parietal regions contributed most strongly to the classification of left versus right decisions. We now describe this topography in the main text (page 11):

“Here we used whole-brain classification techniques to derive efficient predictions of participants’ decisions from across the full MEG sensor array. We asked next whether specific sensor clusters drive the classifier performance. A previous literature using EEG has reported a centro-parietal event-related potential (the centroparietal positivity or CPP) as a neural marker of internal evidence accumulation^{12,30,31}. Accordingly, when identifying the features that contributed most strongly to classifier decoding accuracy (Figure 3C) we found also that centroparietal sensors made a disproportionate contribution to an ability to differentiate between left and right decisions.”

4. While we cannot rule out that early visual (e.g. MT) signals contribute to the classification of participants’ choices, we think that several features of our data are more consistent with

decoding of a downstream decision signal. First, the topography is more consistent with the CPP abstract evidence accumulation signal, as covered in the previous point. Second, our classification accuracy for left versus right decisions reached the highest accuracy around 550-600ms after stimulus onset (Figure S9A), which was 200-250ms after the stimuli disappeared from the screen. Therefore, it seems unlikely that our classifier was only utilizing stimulus-driven activity to decode left versus right choices.

- As the Reviewer notes we designed the task such that participants were not able to prepare their button press in advance of knowing the response mappings. In addition, the responses were all made using the right thumb (pressing an up or down button on a video game keypad), such that differences in movement preparation were also minimised. Nevertheless, we now confirm that movement-related beta activity did not differentiate between left and right decisions (see Figure R1 below). We plot the contrast of broadband power between left and right decisions, separately for the left and right motor cortex (we averaged data from the 12 lateralised channels used by Donner et al. 2009). In accordance with the limitations on motor preparation we found no modulation (i.e. no cluster surviving correction for multiple comparisons) in beta-band power (12-36 Hz) as a function of the decision a participant is going to make.

Figure R1 Beta-band power contrast between left and right decisions. We present the contrast in broadband activity between trials in which participants chose left motion versus right motion. Data is plotted separately for activity in the left motor cortex (left panel), the right motor cortex (middle panel) and the difference between both sides (right panel). We obtained no consistent modulation of beta-band power (12-36 Hz) separating left and right choices during the time of our decoding analysis (0-850 ms).

For comparison purposes please see the following Figure reproduced from Donner et al. (2009), which demonstrates a clear beta-band suppression between stimulus offset and response:

Reviewer #2 (Remarks to the Author):

The work explores the role of decision confidence as a potential driver of confirmation bias. More specifically, the authors investigate how confidence following a primary decision might affect the process of evidence accumulation on a subsequent (secondary) decision. They show that following highly confident choices the integration of confirmatory evidence in the secondary decision is amplified, while disconfirmatory evidence processing is minimized.

The topic is interesting and the paper could potentially add to recent evidence demonstrating how sequential trial dependencies (including confirmation biases) might be the product of confidence-weighted accumulation of relevant evidence (e.g. Urai et al., eLife 2019, Talluri et al., Curr Biol 2018). I do, however, have a number of conceptual and methodological concerns that the authors should consider (see below).

As a general remark, I found it challenging to follow the paper given the current (heavily condensed) format, presumably reminiscent of an earlier submission to a different journal. The paper can benefit from a proper introduction to better motivate the work and the proposed hypothesis. Similarly, results should be presented (and discussed) in sufficient detail to allow the reader to fully digest the main contributions of the work (I would also move some of the supplementary figures in the main text to facilitate this).

Point 5

We thank the Reviewer for a positive overall evaluation. We agree with the above views and have now extended the introduction (page 3, see also point 1), results and discussion to document more fully our methods and findings. Specifically, we present our results in greater detail, including the drift-diffusion modelling (page 6-8) and MEG analysis (page 10-14), as well as extending on the discussion (page 16-17). We trust that these changes render the revised manuscript more accessible and easier to digest.

All changes have been highlighted in the revised version of the manuscript.

In the same spirit, excluding subjects that did not exhibit a “positive evidence” manipulation on changes of mind is misleading, since it conceals inter-individual differences and inflates the generality of the presented findings (there was in fact no statistically significant effect of the manipulation when

all subjects were included; supplementary figure 2). It might be more appropriate to openly present the full extent of the data in the main text.

Point 6

We thank the Reviewer for this suggestion. We agree it is useful to present explicitly individual differences in positive evidence manipulation. We have replaced the previous Figure 1D with Figure S2B from the supplementary material (page 6) and modified the text accordingly:

“We next set out to test whether this boost in confidence influenced changes of mind. There were notable individual differences in the degree to which our manipulation boosted participants’ confidence (see Figure 1D). Importantly, subjects who experienced a stronger confidence boost through the positive evidence manipulation also showed a stronger reduction in changes of mind ($r=-.69$, $p<.0001$, see Figure 1D), an effect not explained by an impact of positive evidence on accuracy or reaction time (p -values $=.005$ when controlling for these effects). This supports a notion that confidence is a driver of reductions in changes of mind (see Supplementary Information for additional behavioural and MEG analyses that further confirm confidence as a critical driver of changes of mind).”

The proposed modelling framework appears to suggest that even following low confidence trials, subjects are more likely to move their starting points (and/or drift speed) towards the initially chosen option (as in illustration of Fig. 2A) – though to a lesser extent than following more confident choices. This is a bit counterintuitive. One might expect that confirmation bias is mainly present for confident trials, while following less confident choices, starting point/drift speed might be more neutral (or even biased in the opposite direction). Have the authors explicitly tested for this possibility? Though I understood the broader model family definitions, I’m still unsure (after reading the methods and supplement) what the exact properties of the 10 different model parameterizations were.

Point 7

We agree with the Reviewer’s intuition that a shift in starting point and drift rate should mainly occur after decisions held with high confidence. Indeed, this is exactly what we show in the interaction term for confidence \times initial decision. However, our winning model also included main effects of the initial decision on starting point and drift rate (see Figure 2D middle panel), as well as this interaction term, and we also find that these main effects are significant. We interpret this as reflecting a general tendency towards confirmation bias that is minimised, but not abolished, following low confidence decisions, a conclusion also consistent with our MEG data. In other words, even if a participant has relatively low confidence their decision can still influence post-decision evidence accumulation (though to a smaller extent). To avoid any confusion on this issue we now explain the drift-diffusion modelling more extensively in the main text (page 6-7):

“We considered two potential mechanisms through which confidence might reduce changes of mind. First, confidence might reflect a shift of the starting point of post-decision accumulation closer to the bound associated with an initial decision, consistent with a continuation of pre-decisional evidence accumulation (influence on starting point; Figure 2A upper panel). Second, confidence may induce selective accumulation of evidence in line with an initial decision (influence on drift rate; Figure 2A lower panel) – a clear instance of confirmation bias.

Critically, these two mechanisms make different predictions in terms of the distributions of response times for the final decision^{8,9}. We compared 10 drift-diffusion models that embodied these different predictions (see Supplementary Information for a full model comparison). We employed accuracy coding such that the bounds correspond to a correct versus an incorrect decision respectively, while a positive drift-rate represents stronger integration of the presented (correct) motion direction. Note, by design, confirmatory post-decision evidence was received when the initial decision was correct, and disconfirmatory evidence when the initial decision was incorrect (Figure 2B-D). In addition, in light of suggestions that confidence might also affect the separation of decision bounds, and thus the trade-off between speed and accuracy of subsequent decisions^{28,29}, we allowed also for a dependency of boundary separation on initial confidence in all models.

These models differed in whether the starting point and/or drift-rate were affected by confidence (models 2-4), accuracy of the initial decision (models 5-7; i.e. correct=1 and incorrect=-1, capturing a general confirmation bias) and their interaction (models 8-10; i.e. capturing a confirmation bias that depends on confidence). The winning model (Model 10 based on the DIC score, see Figure S3A) incorporated dependencies of starting point and drift-rate on all factors (confidence, initial decision and the interaction) and provided a good fit to the data (Figure 2B&C):

starting-point $\sim 1 + \text{confidence} + \text{initial decision} + \text{confidence} \times \text{initial decision}$
drift-rate $\sim 1 + \text{post-decision evidence strength} + \text{confidence} + \text{initial decision} + \text{confidence} \times \text{initial decision}$
boundary separation $\sim 1 + \text{confidence}$ ”

We also now include more details on the parameterization of each model family in Methods (page 26-27):

“First a baseline model was estimated where none of the parameters depended on confidence or an initial decision. Subsequently, we created three model families that had dependencies of starting point and/or drift-rate on (i) initial confidence, (ii) initial decision or (iii) the interaction of initial confidence \times initial decision (i.e. confidence was allowed to amplify or attenuate the influence of the initial decision on the starting point and/or drift-rate). Within each model family we created three different models with dependencies of these variables on starting point, drift-rate or both:

Baseline model (Model 1):

starting point ~ 1
drift rate $\sim 1 + \text{post-decision evidence strength}$
boundary separation $\sim 1 + \text{confidence}$

Confidence dependency (Model 4):

starting point $\sim 1 + \text{confidence}$
drift rate $\sim 1 + \text{post-decision evidence strength} + \text{confidence}$
boundary separation $\sim 1 + \text{confidence}$

Initial decision dependency (Model 7):

starting point $\sim 1 + \text{initial decision}$
drift rate $\sim 1 + \text{post-decision evidence strength} + \text{initial decision}$
boundary separation $\sim 1 + \text{confidence}$

Full model (Model 10):

starting point $\sim 1 + \text{confidence} + \text{initial decision} + \text{confidence} \times \text{initial decision}$
drift rate $\sim 1 + \text{post-decision evidence strength} + \text{confidence} + \text{initial decision} + \text{confidence} \times \text{initial decision}$
boundary separation $\sim 1 + \text{confidence}$ ”

In this spirit, I believe it would be useful to further divide up RTs based on the accuracy of the initial choice (as was done in Fig. 2B for accuracy). For example, in the alternative formulation proposed above, when subjects are highly unsure in their primary decision, the relative speed with which they repeat or reverse their choice in the secondary decision could be different than the current model’s prediction.

Point 8

We thank the Reviewer for raising an interesting issue. We have performed this analysis, separating RTs for initially correct and incorrect trials as a function of the initial confidence and whether participants changed their mind (see Figure R3 below). Importantly, our model continues to capture

these qualitative patterns even when the data are split further. However, it is noteworthy that the fit is poor for a subset of conditions which contain small numbers of trials (e.g. for some participants there were only 2 trials with changes of mind after an initially correct decision, and the group median was 12 trials per participant), likely driven by the lack of precision in both the measurement and model simulations. We now include Figure R3 in Supplementary Information (Figure S8)

Figure R3 Model simulations (of the best fitting model) reproduce behavioural patterns of reaction times of the second decision. Data is presented separately for initially correct decisions (left panel) and initially incorrect decisions (right panel). Model simulations are shown as dotted lines, behavioral data as solid markers. Error bars indicate \pm 95% confidence interval.

Following from this, in addition to the formal model comparisons it would be important to quantify the goodness of fit of the winning model (R^2 or similar). In other words, the winning model might still be a somewhat poor fit to the data. Visualizing the average RT fits as in Fig. 2C, masks the extent to which the model captures the full profile of the RT distributions (skewness, heavy tails etc). It would be more informative if the full RT distributions from the data and model simulations were superimposed (separately for each of the relevant conditions) to allow a comprehensive comparison.

Point 9

We thank the Reviewer for this suggestion. We followed the Reviewer's advice and include an additional panel of Figure 2C showing the full distribution of the empirical RTs and respective model predictions for each of the different conditions. We think this now makes it possible to appreciate both the qualitative different patterns between the different conditions and allow an appreciation of overall model fit. We also calculated the chi-square statistic to quantify the goodness of fit, as suggested by Ratcliff and Tuerlinckx (2002, Psychonomic Bulletin & Review), returning a $\chi^2=648.29$.

Since it has been suggested that confidence might influence the boundary separation of subsequent decisions (Desender et al., 2019, eLife; Van den Berg et al., 2016, Current Biology), we also explored the possibility that initial confidence might influence the boundary separation, and thus a speed-accuracy trade off. After plotting the full reaction time distributions against the predictions of our

model, we realized that incorporating a dependency of boundary separation on initial confidence might provide a better fit to the data. We now allow such a dependency in all models, and all reported effects remain significant when controlling for this. We include this in the revised manuscript (page 7):

“In addition, in light of suggestions that confidence might also affect the separation of decision bounds, and thus the trade-off between speed and accuracy of subsequent decisions^{28,29} we also allowed for a dependency of the boundary separation on initial confidence in all models.”

*Regarding the MEG analysis, I'm a bit surprised that one could discriminate the choice (left vs right). This implies that there exist separate accumulators for left and rightward choices in *spatially* distinct regions such that the MEG can resolve them – this seems unlikely based on the architecture of known decision accumulator regions. The other possibility is that the classifier exploits the encoding of left/right evidence in sensory cortex, which also seems unlikely given that different motion directions are encoded within the same area(s) (MT/MST) and MEG would not be able to resolve them. One thing that is unclear from the methods is whether subjects used the same hand to indicate their choice or whether different choices were mapped onto separate hands, which can have implications on how these signals are interpreted.*

Point 10

We agree that it is a non-trivial exercise to decode an abstract left versus right choice from MEG data. While spatial proximity of different accumulators might render it difficult to detect activity differences in a univariate analysis, we believe our multivariate decoding approach provides us with the statistical power needed to pick up subtle, but systematic, differences in activity distributed over the whole scalp. Note that this need not imply that anatomically distinct regions encode left and rightward accumulators – instead, as for example seen with orientation decoding in V1, subregional variation in the coding of decision variables may lead to detectable biases in sensor-level activity profiles. In keeping with this interpretation, the sensors that maximally contributed to the decidability of choices (see new Figure 3C) were located to a centro-parietal cluster corresponding to brain region that usually are associated with abstract evidence accumulation. However, while our results are in line with previous findings on neural evidence accumulation signals, our multivariate approach enabled a more flexible analysis of MEG data. The neural basis of abstract evidence accumulation remains largely unknown and these accumulators may not be restricted to visual, centroparietal or motor areas.

We apologise if our methods were unclear. In our task, participants responded always with their right thumb to press either an up or down button on a keypad. Moreover, the mapping between responses and button presses was randomized for every decision so it is unlikely a classifier was picking up on any motor preparation signals (see also point 4 in response to Reviewer 1 for further analysis that pertains to this issue). We now include a sentence in the methods to explain the way responses were indicated (page 22):

“In the MEG study, participants indicated their responses by pressing and up or down button on a keypad with their right thumb”

One thing that is missing from the paper and could potentially resolve some of these issues are the scalp topographies associated with the signal extracted from the classifier. Since the work uses linear SVMs it would be trivial to estimate a forward model to see how the extracted signal projects back onto the sensors. The process of evidence accumulation has a very prototypical scalp topography as shown in a number of papers from several groups (centroparietal positivity). This could further validate whether the observed signal is indeed a reliable metric for the process of evidence accumulation.

Point 11

We thank the Reviewer for this excellent suggestion, also raised by the other reviewers. Due to shared variance between different MEG channels an interpretation of SVM weights is not straightforward (Haufe et al., 2014 NeuroImage). Instead, to explore which brain areas carried information for the left

versus a right decision, we again trained an SVM - but instead of using all 273 sensors, we repeated the analysis 2500 times using random subsets of 30 sensors. The contribution of each sensor s was taken to be the mean of all prediction accuracies achieved using the ensembles of 30 sensors that included s . This approach breaks the covariance between the different sensors and allows an evaluation of the contribution of each sensor to decoding accuracy (as also used by Liu et al., 2019 Cell, and Kurth-Nelson et al., 2015 eLife).

In line with the Reviewer's suggestion, this analysis validated a prototypical scalp topography in centro-parietal regions as being the most relevant brain areas for decoding left versus right decisions (see new Figure 3C).

We now include a Figure of this scalp topography (Figure 3C) and elaborate on these findings (page 11):

“Here we used whole-brain classification techniques to derive efficient predictions of participants' decisions from across the full MEG sensor array. We asked next whether specific sensor clusters drive the classifier performance. A previous literature using EEG has reported a centro-parietal event-related potential (the centroparietal positivity or CPP) as a neural marker of internal evidence accumulation^{12,30,31}. Accordingly, when identifying the features that contributed most strongly to classifier decoding accuracy (Figure 3C) we found also that centroparietal sensors made a disproportionate contribution to an ability to differentiate between left and right decisions.”

Another observation that casts some doubt on the interpretation of the proposed accumulator signals is the negative going slopes on some conditions. Why would this signal “ramp down”? Unlike single neuron response profiles (as in LIP for example) coding for specific choices, MEG signals reflecting populations responses over groups of neurons are unlikely to exhibit such selectivity. This point is also tightly linked to my earlier comment regarding the origin and nature of the discriminating activity.

Point 12

We apologise that this analysis was unclear. The ramping down of the signal in some conditions is simply due to the way the classifier is set up and coded. A negative value indicates a classifier prediction for a left decision due to how we coded the variables (1=right decision, -1=left decision, see Figure 3B in the revised manuscript). We have now made this clear on page 10:

“We then applied the trained classifier to brain activity at the corresponding time point in the post-decision time window, enabling us to derive a probabilistic prediction of internal evidence favouring a leftward versus rightward decision (see Figure 3A left panel), where the strength of this prediction indexes neural evidence in favour for the respective decision. Positive values indicate prediction of a rightward decision and negative values indicate prediction of a leftward decision (see Figure 3B). We next fitted a linear regression to the time series of classifier predictions within each trial (see Figure 3A right panel) to obtain a trial-by-trial neural measure of the starting point (intercept) and drift rate (slope). These measures of evidence accumulation (slope) should be highly responsive to the presented motion direction during the post-decision period, and we show this was indeed the case (hierarchical regression: $\beta = .08$, $p < 10^{-14}$, Figure 3B). “

However, these predictions represent a stimulus dependent measure of evidence accumulation, i.e. positive values indicate integration of evidence for rightward motion and negative values for leftward motion. In order to obtain a stimulus independent metric of the veridicality with which the actual motion direction was integrated, we flipped the sign of the slopes for trials in which leftward motion was presented. This transformation quantifies the propensity to correctly integrate the presented information, by tracking an internal decision variable (analogous to a drift rate in an accuracy coded drift-diffusion model as used in Figure 2), where higher values indicate stronger incorporation of the objective stimulus information. In other words, a positive value in our MEG decoding analysis (Figure

4A-C) indicates neural evidence in favour of the actual (correct) motion direction, whereas a negative value indicates neural evidence favouring the wrong motion direction. We clarify this now in the main text (page 10):

“The slopes extracted from this analysis are signed, such that positive values indicate stronger prediction of a rightward choice and negative values stronger evidence for a leftward choice. In order to obtain an unsigned metric of evidence accumulation strength, we flipped the sign of the slopes extracted from trials in which leftward motion was presented (we conducted the same flip for the intercept to obtain an unsigned metric of the starting point). This measure quantifies the propensity to correctly integrate the presented information, by tracking an internal decision variable (analogous to a drift rate in an accuracy coded drift-diffusion model as used in Figure 2), where higher values indicate stronger sensitivity to the presented stimulus.”

On a more methodological note, the approach for estimating slopes assumes an one-to-one mapping between pre- and post-decision periods, but the way decision dynamics unfold (e.g. onset of accumulation) might be different across the two periods. For example, the hot spot above main diagonal in fig 3A, suggests the generalization in the post-decision period happens earlier. An alternative approach would be to identify individual peak generalization performance, even if the relevant time point appears off the main diagonal, and work backwards along that diagonal (i.e. follow a new trajectory).

There seems to be a discrepancy between the model and the MEG predictions on starting point? Is this an issue with the modeling or the extraction of the relevant MEG signals? Please note that starting point manipulations (e.g. changes in prior beliefs might manifest more in the frequency content of the signal rather than the evoked responses on which the classifier was trained on).

Point 13

We thank the Reviewer for raising these astute points. While it is possible that post-decision processing stages are systematically shifted with respect to corresponding pre-decision stages, in general we think it is striking that a generalization of pre- to post-decision time points is nicely aligned along the major diagonal. In contrast, early in the trial there appears to be a fundamental asymmetry between the starting point of evidence accumulation in the pre-decision phase (where no evidence is yet accumulated) and the post-decision phase (where evidence has already been accumulated). This putative asymmetry in starting point was a key motivation for conducting the temporal generalization analysis in Figure 3E. Importantly, this analysis does not assume a systematic mapping between pre- and post-decision dynamics but makes it possible to detect whether processing stages during the pre-decision phase reappear at different times in the post-decision phase, and how this change in timing is modulated by confidence. Such an analysis is well suited to account for potential differences in a mapping between pre- and post-decision processing stages highlighted by the reviewer.

Interestingly, very early in the post-decision period a reinstatement of later processing stages of the pre-decision period occurs when participants are highly confident. This finding is in line with confidence shifting a starting point towards the initial decision. We now motivate this analysis in more detail in the revised manuscript (page 14):

“We further reasoned that this approach may remain blind to changes in the starting point of post-decision evidence accumulation because of an asymmetry in evidence availability at the start of the pre- and post-decision phases. In other words, reapplying the (non-predictive) classifier weights obtained from the beginning of the pre-decision phase to the same time point in the post-decision phase could render it impossible to identify a starting point offset towards a left or right response. To address this issue, we evaluated the extent to which the entire timecourse of classifier predictions obtained in the pre-decision phase generalised to the post-decision phase, without making assumptions about their relative phase³². This analysis provides insight into how putative processing stages identified in the pre-decision phase are reinstated in the post-decision phase, and crucially how this timecourse is affected by confidence. We found a cluster of time points in which a representation

of the initial decision was activated earlier in the post- compared to the pre-decision phase when confidence was high ($p=.01$, corrected for multiple comparisons; Figure 3E). Such early reinstatement of a later processing stage is consistent with confidence enhancing a representation of the initial decision (i.e. shifting a starting point towards the bound of the initial decision) or inducing an expectation for evidence supporting an initial decision at the beginning of the post-decision period. Together these results indicated that confidence changes both the neural representation of evidence for an initial decision at the beginning of the post-decision phase (analogous to a change in starting point), as well as enhancing the processing of evidence supporting an initial decision (analogous to a change in drift rate)."

The reviewer raised an interesting point that shifts in starting point might also manifest in frequency content as well as evoked responses. Because we had limited hypotheses about these (potentially high-dimensional) features of the data we would prefer to defer the exploration of effects of confidence on such oscillations to future work.

*If understood the analysis correctly, Fig. 3A shows results whereby training was done on pre-decision data and the classification weights were used to predict the secondary choice on post-decision data, whereas in Fig. 3E the same procedure was adopted to predict the *initial* choice on post-decision data. Assuming that's true, do the effects from 3E predict the slopes in the analysis presented in 3A? In other words, the effects presented in 3E can be seen as a proxy of the initial influence of confidence on the secondary decision.*

Point 14

The Reviewer picks up an interesting and subtle point here. The classifier weights are always trained on pre-decision MEG data in order to predict the initial decision made at the end of the pre-decision period (this is the case both for Figure 3A and 3E). These classifier weights are then applied to decode post-decision MEG data (both for the analysis in Figure 3A and also for Figure 3E). Based on the classifier weights and the post-decision MEG data, the classifier always gives a (probabilistic) prediction and this prediction can be seen as decodable neural evidence for a left versus right decision. For the analysis in Figure 3A we extract directly the change in this prediction within each trial. For the analysis in Figure 3E we quantify a decoding accuracy by comparing this prediction to the actual decision (in this case the first decision). Therefore, the Reviewer's intuition is correct in that the analyses of Figure 3A and 3E both rely on the same classifier predictions. However, there are important differences between these two analyses. While the analysis in Figure 3E returns an average decoding accuracy for each participant at each timepoint, the analysis in Figure 3A provides a trial-by-trial measure of the slope of neural evidence accumulation (by fitting a regression to the predictions obtained across timepoints within each trial). In contrast, Figure 3E does not allow a direct inference of how evidence accumulation changes within each trial, as in this analysis each time point is analysed independently of all other time points, and different subsets of trials could contribute to the decodability observed at different time points.

Are the 2D images shown in fig. 3A, E from individual subjects or the population? For Fig. 3E, which of the four trial types does this panel correspond to? What happens to the other conditions? For example, are there any interesting dissociations that can further validate the proposed effects?

Point 15

The images in Figure 3A&E are averages over the whole population (we now make this explicit in the legend). Figure 3E is a main effect of confidence on the representation of the initial decision during the post-decision period, averaged over all conditions. For this main effect we calculated the contrast of high > low confidence averaging over change/no change of mind trials [[high confidence & no change of mind - low confidence & no change of mind] + [high confidence & change of mind - low confidence & change of mind]].

We found that an enhanced representation of the initial decision when subjects were highly confident was present independently of whether participants change their mind later on or not, suggesting that this effect does not dissociate meaningfully across conditions (however we note that caution is needed here as the high-confidence change-of-mind condition contains a limited number of trials).

Minor:

I would tone down the examples given in the abstract (and at the beginning of the paper) regarding the utility of the observed effects in complex societal issues. The task used here involves a simple perceptual discrimination task and it remains unclear whether the reported effects (and underlying mechanisms) extend to other types of decisions involving great emotional and cognitive influences.

Point 16

We agree with the Reviewer that more caution is needed here and have toned down the examples used in the abstract and introduction (see also point 1 in response to Reviewer 1).

Reviewer #3 (Remarks to the Author):

In this work Rollwage and colleagues report the result of three studies showing that high confidence in a first decision induces a confirmation bias in a second (linked) decision. Using drift diffusion modeling they demonstrate that confidence affects the starting point and drift rate of the second decision. Using MEG recordings, they show that after high confidence decisions choice inconsistent evidence is largely ignored. I think this is an interesting piece of work, with also quite elaborated and consistent additional analyses reported in the supplements. Below, I outline a couple of reservations I had, mostly related to the strength of some claims, fitting procedures and overall clarity.

We thank the Reviewer for a positive assessment and have addressed their very helpful comments in full below.

Major comments

1. The effect of confidence on the drift rate in Experiment 2 was confusing. Traditionally, the DDM is represented with bounds for left versus right responses. With such implementation, a confirmation bias would be expressed as a stimulus-independent (but previous choice dependent) constant added to the mean drift (“a drift criterion”). However, the authors instead model the two bounds as chosen versus unchosen, so that the confirmation bias boils down to a simple drift rate effect. I think explaining this more carefully would help to avoid quite some confusion. In the methods, it is stated that accuracy coding was used. However, if initially the wrong option is selected with high confidence, a higher drift rate implies a higher probability of ending up at the correct bound (whereas the data shows less changes of mind with high confidence).

Point 17

We thank the Reviewer for prompting further clarity. As the Reviewer points out we used accuracy coded DDMs such that the drift-rate corresponds to the veridicality with which the presented evidence is integrated (boosting the chances of a correct decision). Thus, a confirmation bias leads to a positive influence on drift rate after correct decisions (the participant is more likely repeat a correct decision), and a negative influence on drift rate after incorrect decisions (the participant is more likely to repeat a wrong decision). Importantly, we model the interaction between confidence and initial accuracy when seeking to account for a confidence-driven confirmation bias. Therefore, after initial incorrect decisions made with high confidence, the drift rate (and starting point) is reduced and there is a higher probability to repeat an incorrect decision. In the revised manuscript we elaborate more on our model specification and how to interpret the results in order to avoid confusion (page 7-8):

“After accounting for main effects, we observed a dependency of the starting point on the interaction between confidence and initial decision (95% equal-tailed interval=.08-.18; Figure 2D righthand panel), indicating participants started the accumulation process closer to the bound of the initial decision when highly confident in their choice. Even more striking was the discovery of a similar interaction effect on drift rate (95% equal-tailed interval=.11-.26; Figure 2D righthand panel) indicating participants selectively accumulated evidence supporting their initial choice, and were more likely to do so when they were more confident. Such a confirmation bias led to a boost in accumulation of the veridical motion direction following high-confidence correct decisions (as such information served to confirm the original choice), whereas it led to a reduction in evidence accumulation (manifest as a lowered drift rate) following high-confidence errors (as new information served to disconfirm their originally wrong decision).”

2. I am a bit worried about the fitting procedure of the DDM. The posterior distributions of the DDM fits shown in 2D look quite noisy, and are based on a rather low number of posterior samples (1000, which is only a fraction of what is done usually). The very large burn-in and thinning seem to suggest difficulties with the fitting procedure. It would be good if the authors could demonstrate model convergence formally (i.e., by computing the Gelman-Rubin statistic), show the autocorrelation patterns, and provide any other relevant information that can help to alleviate any concerns there.

Point 18

We thank the reviewer for this helpful suggestion. Since our family of regression models included higher-order interaction terms we employed a long burn-in and substantial thinning to be on the safe side regarding convergence of the sampler. This approach ensured successful convergence: as we now report in the revised manuscript, the Gelman-Rubin statistic was close to 1.00 for all model parameters (see page 6 of Supplementary Information). To check our results were not an artefact of undersampling we also explored the impact of increasing the draws of the posterior samples to 2500 and find exactly the same results as beforehand. Please see also the autocorrelation patterns of our chains to show that indeed all our chains have successfully converged.

starting point ~ confidence

drift-rate ~ confidence

starting point ~ initial decision

drift-rate ~ initial decision

starting point ~ confidence x initial decision

drift-rate ~ confidence x initial decision

Figure R4 Traces, autocorrelation and histograms of the marginal posterior of parameter draws for each model parameter of the best fitting model.

3. The authors make a very strong claim about showing a “causal” effect of confidence on post-decision processing. The findings are certainly in line with an explanation in such terms, but I think a bit more caution is warranted, and some claims about causality should be toned down a bit. For example, the effect of confidence in Figure S6B might be due to any other variable jointly influencing confidence and changes of mind.

Point 19

We thank the Reviewer for prompting further reflection on this issue and have toned down our claims about causality.

4. Although I thought the MEG analyses very elegantly demonstrated a neural confirmation bias, they were a bit of a “black-box”. We know quite a lot about the neural implementation of perceptual decision making (e.g., CPP, motor beta), and not so much yet about neural implementation of confidence. It would be informative to provide some more insight about what the decoder is picking up on. For example, scalp distributions might be useful to see whether the choice decoder is “just” picking up on something like the CPP or motor beta, or whether something else is going on. The same goes for the decoding of confidence.

Point 20

We thank the reviewer for their enthusiasm for our MEG approach and agree that it is informative to investigate the topography of signals contributing to the decoder (see also point 4, 10 & 11). In our new Figure 3C we show the choice decoder is largely dependent on predictions arising from centro-parietal regions with a similarly topography to the CPP. As suggested by the Reviewer we have also investigated the scalp topographies of the confidence classifier and report this in Supplementary Information (Figure S6C&D). Interestingly, the topography for confidence classifier appears to be more distributed than the choice classifier, with a greater contribution from fronto-central areas, but also a noticeable contribution of occipital areas. We note that this occipital contribution might reflect a top-down effect given that the maximum decodability of confidence is obtained at a time point when the stimulus is no longer presented on the screen. We discuss those topographies of the confidence classifier now in the Supplement (Supplement page 13-14):

“Moreover, investigating the topography of features (i.e. channel activity) that contributed to the decoding of confidence and choices, revealed that these two classifiers picked up distinct brain activity patterns (see Figure S6 C&D). The classification algorithm that predicted left versus right decisions mainly utilized activity in centro-parietal regions to predict participants upcoming decisions (see Figure S6 D). In contrast, the confidence classifier utilized patterns of central and frontal brain activity as well as occipital regions (see Figure S6 C). Note however that the most marked occipital contribution to the confidence classification was apparent at late time points of the trial, long after the stimuli disappeared from the screen (the dots were only shown for 350 ms), indicating that this contribution is unlikely to simply represent a feature of the stimuli per se, but rather a potential top-down effect on occipital regions“

Minor comments

5. For some temporal generalization matrices, it is unclear whether stats are missing or whether there are no significant clusters (e.g., Figure S8A and S9). This should be clear.

Point 21

We thank the Reviewer for spotting this. For the mentioned Figures we have not conducted statistical tests as these serve descriptive and illustrative purposes, and this is now mentioned in the Figure legends.

6. Because much of the introduction focused on evidence accumulation models, it was confusing that Figure 1 displays a design which –on the look of it- does not feature two-choice responses nor speeded response, both of which are essential for fitting the DDM. It would be good if the information that it is in fact a 2CRT task is not hidden in the methods.

Point 22

We thank the Reviewer for spotting this omission – we include now in the legend of Figure 1 that this is a classical 2AFC task.

7. I might have missed it, but did Experiments 2 and 3 also show an increase in confidence and a decrease in changes of mind for HPE? In the main text, only significant mediation in all three experiments is mentioned.

Point 23

Yes, in Experiments 2 and 3 we replicate that an increase in confidence under HPE (experiment 2: $p=.016$; experiment 3: $p=.003$) is strongly linked to the degree to which the manipulation affected changes of mind (experiment 2: $r=-.53$, $p=.009$; experiment 3: $r=-.71$, $p<.0001$; see point 6 regarding the update in how we describe our analysis strategy suggested by reviewer 2).

8. The colors in Figure 2 were ambiguous. Why are different colors used to in panel A to demonstrate the effect of starting point versus drift rate? The same colors appear in panel B without any relation, and again in panel C but now the mapping is even reversed compared to A.

Point 24

We are not entirely sure what the Reviewer is referring to regarding the flip in colour. In all the images in Figure 2, green is used to indicate an effect on the starting point (illustrated in figure 2A upper panel) and this is consistently carried over to Figure 2D where the green posterior distributions represent the effects of confidence (left panel), initial decision (middle panel) and their interaction on the starting point. Similarly, violet is used to illustrate effects on the drift rate (illustrated in Figure 2A lower panel) and this colour code is also applied to Figure 2D where the violet distributions represent the dependencies of the drift-rate on confidence (left panel), initial decision (middle panel) and the interaction (right panel). We hope this clarifies our colouring and would be glad to revisit how we present these results if the Reviewer wishes us to do so.

***REVIEWERS' COMMENTS:

Reviewer #1 (Remarks to the Author):

The authors have provided a very comprehensive and, in my view, satisfactory response to all the comments raised and I am happy to recommend acceptance for publication

Reviewer #2 (Remarks to the Author):

The authors have adequately addressed my initial concerns. The manuscript reads a lot better and there is now sufficient detail to follow the various analyses as well as the interpretation of the main results.

Reviewer #3 (Remarks to the Author):

In this revision, Rollwage and colleagues did a great job at carefully addressing all the points I previously raised. I also had another careful look at the manuscript itself, which has substantially improved compared to the previous version.

I do not have any further comments, and want to congratulate the authors on a very fine piece of work.

Reviewer #1 (Remarks to the Author):

The authors have provided a very comprehensive and, in my view, satisfactory response to all the comments raised and I am happy to recommend acceptance for publication

Reviewer #2 (Remarks to the Author):

The authors have adequately addressed my initial concerns. The manuscript reads a lot better and there is now sufficient detail to follow the various analyses as well as the interpretation of the main results.

Reviewer #3 (Remarks to the Author):

In this revision, Rollwage and colleagues did a great job at carefully addressing all the points I previously raised. I also had another careful look at the manuscript itself, which has substantially improved compared to the previous version.

We thank all reviewers for their time spent on the review process, which we think has greatly strengthened the final paper.